# MEASURING NON-ADVERSARIAL REPRODUCTION OF TRAINING DATA IN LARGE LANGUAGE MODELS

**Michael Aerni**[*1]   **Javier Rando**[*1]   **Edoardo Debenedetti**[1]
**Nicholas Carlini**[2]   **Daphne Ippolito**[2,3]   **Florian Tramèr**[1]

[1]ETH Zurich   [2]Google DeepMind   [3]Carnegie Mellon University

## ABSTRACT

Large language models *memorize* parts of their training data. Memorizing short snippets and facts is required to answer questions about the world and to be fluent in any language. But models have also been shown to reproduce long verbatim sequences of memorized text when prompted by a motivated adversary. In this work, we investigate an intermediate regime of memorization that we call *non-adversarial reproduction*, where we quantify the overlap between model responses and pretraining data when responding to *natural and benign prompts*. For a variety of innocuous prompt categories (e.g., writing a letter or a tutorial), we show that up to 15% of the text output by popular conversational language models overlaps with snippets from the Internet. In worst cases, we find generations where 100% of the content can be found exactly online. For the same tasks, we find that human-written text has far less overlap with Internet data. We further study whether prompting strategies can close this reproduction gap between models and humans. While appropriate prompting can reduce non-adversarial reproduction on average, we find that mitigating worst-case reproduction of training data requires stronger defenses—even for benign interactions.

## 1 INTRODUCTION

Large language models (LLMs) must memorize parts of their training data, including facts and idioms, to generate fluent text and answer questions about the world. The rate at which LLMs memorize atomic facts or word constructs (e.g., small ngrams) is measured by general knowledge benchmarks (Hendrycks et al., 2020) and studies of linguistic novelty in LLMs (McCoy et al., 2023; Nguyen, 2024; Lu et al., 2024). While this form of memorization is desired and necessary, models have also been shown to memorize long sequences of verbatim text that can be extracted by motivated adversaries (Carlini et al., 2021; Nasr et al., 2023).

In this paper, we consider an intermediate regime and measure *non-adversarial reproduction*, that is, the extent to which an LLM's outputs overlap with the public content of the Internet[1] when answering *natural prompts* in standard *benign* situations. This regime thus interpolates between the two previously studied extreme forms of LLM memorization, i.e., natural reproduction of short ngrams and adversarial extraction of large verbatim texts. Concretely, we collect outputs from state-of-the-art conversational LLMs prompted with a variety of common and benign tasks (including real conversations from WildChat (Zhao et al., 2024) and LMSYS-Chat-1M (Zheng et al., 2023)). We then measure the fraction of generated text that overlaps (to varying degrees) with snippets of text from the public Web, and compare this with human-written baselines for the same tasks.[2]

Our results show that, even in benign settings, the outputs of production conversational LLMs routinely contain text snippets from the Internet (see Figure 1 for an illustration). On average, 8–15% of the text generated by LLMs overlaps with strings of at least 50 characters that appear verbatim online. We find that the rate of such overlaps varies significantly between tasks, with much higher

---

[*]Equal contribution; correspondence to research@michaelaerni.com

[1]We use public internet content as a proxy for the models' (unknown) training data.

[2]Code and data: https://github.com/ethz-spylab/non-adversarial-reproduction.

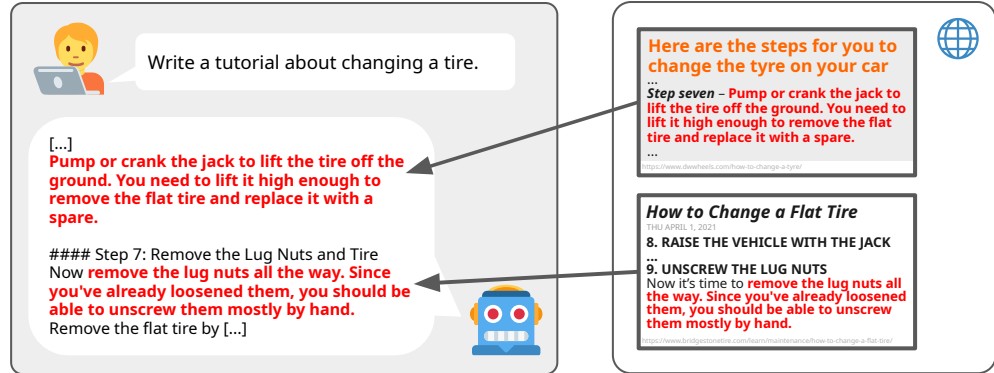

Figure 1: LLMs often output text that overlaps with snippets of their training data when responding to benign prompts. Red text indicates snippets that were found verbatim on the Web.

rates for expository tasks (e.g., "Write a tutorial about setting up an Nginx server.") compared to creative tasks (e.g., "Write a satire about bad coffee."). In fact, the first prompt resulted in the longest reproduced text in our study (see Appendix D.2). Non-adversarial reproduction is long-tailed and in the most extreme cases, we find that models can generate responses where nearly 100% of the content matches existing online text, often by combining snippets from multiple sources.

To distinguish whether overlaps with existing text are due to memorization or simple chance, we compare LLM generations with human-written texts on the same tasks. Our results indicate that, in comparison to humans, LLMs more frequently output moderately long strings found on the Internet. Finally, we study prompting as a possible mitigation for non-adversarial reproduction. Encouraging creativity in the prompt can significantly reduce overlaps with existing text on average but cannot prevent the occasional reproduction of very long sequences.

In summary, our work initiates the study of data reproduction in natural interactions between LLMs and benign users. Our results suggest that LLMs are likely to output sequences of existing text that users may then inadvertently redistribute.

## 2 PRELIMINARIES AND SETUP

### 2.1 MOTIVATION

LLMs retain atomic facts (e.g., "*Paris is the capital of France*") and common idioms (e.g., "*to the best of my understanding*") to answer questions about the world and produce fluent text. However, models also memorize longer sequences that may not be necessary for performance, such as individuals' personal contact information or entire articles from news outlets, that can be extracted through adversarial interaction with the model (Carlini et al., 2021; Nasr et al., 2023). Such long-form memorization raises concerns for privacy and copyright (Grynbaum & Mac, 2023).

The middle ground between these two forms of memorization is yet poorly understood: when does memorization transition from being necessary for language understanding to becoming problematic reproduction of existing content? This question is particularly relevant for moderate-length text snippets that models might reproduce during natural interactions with users. For instance, if a user believes the generation they obtain from a model is novel text—but actually contains fragments copied from existing work (without attribution)—they might face unintended consequences if they redistribute it. Although previous work suggested that training data reproduction is rare in natural usage of *code* generation models (Ziegler, 2021), there is no comparable evaluation of this phenomenon in state-of-the-art conversational LLMs.

Moreover, LLM developers have dismissed claims of unattributed reproduction of third-party content (Grynbaum & Mac, 2023), arguing that adversarial extraction methods violate their usage policies and that data "regurgitation" is otherwise a rare phenomenon (OpenAI, 2024). This position raises important questions about responsibility in cases of unintentional data reproduction. Our work thus measures how often model responses to natural and benign user prompts contain moderate-length snippets of reproduced pretraining data.

## 2.2 METHODS AND EXPERIMENTAL SETUP

**Collecting benign user prompts.** A benign user prompt is an input to a language model system that is designed to accomplish some realistic user task and has not been explicitly designed for the goal of extracting training data. In our analysis, we select three classes of tasks, broadly inspired by Egon (1976): *creative writing* (creative expression), *expository writing* (explain, inform, and describe facts), and *argumentative writing* (compare views, judge, and persuade). To create a diverse set of prompts, we employ several methods:

1. We manually define different tasks and generate corresponding prompts, e.g., "Write a travel blog post about Rwanda.".
2. We collect prompts from real-world sources, e.g., the PERSUADE 2.0 (Crossley et al., 2023) essay corpus or the `r/WritingPrompts` and `r/explainlikeimfive` subreddits.[3]

In total, this yields 3,696 unique prompts over 15 tasks. Further details about prompt construction and examples can be found in Appendix A.1.

Since our prompt dataset is undoubtedly less diverse than actual usage of LLMs, we additionally analyze two publicly available large-scale datasets of real-world LLM conversations. We sample 58,164 conversations from WildChat (Zhao et al., 2024) and 14,675 conversations from LMSYS-Chat-1M (Zheng et al., 2023) to investigate the occurrence of text that can also be found online. For these datasets, rather than running generation ourselves, we analyze the LLM-generated outputs present in the datasets' conversations.

**Defining non-adversarial reproduction.** Nasr et al. (2023) introduce the term *regurgitation* to describe adversarially extracted text that exactly reproduces training examples. We contrast this with *non-adversarial reproduction*, a term we introduce to refer to verbatim reproduction of training data in LLM outputs for benign and natural user prompts. We consider a substring of generated text to be reproduced if it can be found exactly in the training data. Since the real training data of production LLMs is unknown, we use a large fraction of the public Internet as a proxy.

**Measuring non-adversarial reproduction.** Any non-trivial text will inevitably contain some reproduced substrings (e.g., individual characters or words). We hence focus on reproduced substrings of some minimal length, namely at least 50 characters. This threshold is shorter than the 50 *token* (150–200 characters) threshold used in previous studies on adversarial extraction (Carlini et al., 2021; Nasr et al., 2023), but, as can be expected, benign prompting leads to less overall reproduction than adversarial prompting (see, e.g., the tails in Figure 3). Qualitatively, we find that 50-character strings can be both memorized rare strings, as well as very common and unoriginal phrase constructions or idioms. We thus view this as a reasonable interpolation spot between desirable and undesirable memorization. In our analysis, we therefore report two quantities: (1) the proportion of a text that overlaps with such a reproduced substring of length at least 50 characters (we term this quantity the *overlap rate*); and (2) the distribution of the lengths of reproduced substrings. For the latter quantity, we focus on very long reproductions to get a more fine-grained perspective on memorization of rare strings. We report all averages balanced over tasks and text types.

**Filtering prompt snippets and refusals.** In some cases, the prompts we consider may themselves contain snippets of text that can be found on the Web (e.g., "Write an essay about the quote: "The definition of insanity is doing the same thing [...]""). An LLM might then copy such a snippet from the prompt into its output, independent of the snippet's existence on the Internet. We thus discount the length of substrings that were found on the Internet by their longest common substring with the prompt. We explain the exact procedure in Appendix A.2.

Additionally, models sometimes refuse to generate specific content given a benign prompt (e.g., declining to write a book review due to copyright concerns). We use simple heuristics, detailed in Appendix A.3, to filter out API errors, short generations, common refusal prefixes like "I can't assist".

**Establishing a baseline for reproduction in human-written text.** To contextualize our results, we measure how often humans write snippets that would be considered reproductions by our metric

---

[3]We only sample prompts and comments posted after the training cut-off dates of all LLMs we study.

if an LLM were to generate them. To match the text types in our general experiments, we source the following texts as human-written baselines:

- For creative writing, we use 1,000 prompts from the `r/WritingPrompts` subreddit; we compare human-written short stories to LLM generations on the same prompts.

- For argumentative writing, we select the top 250 movies on IMDb (ignoring 8 recent ones that were not included in all LLMs' training data); we compare a total of 4,388 human-written reviews to 3 LLM-generated reviews per movie (positive/negative/unspecified).

- For expository writing, we collect 1,000 questions from the `r/explainlikeimfive` subreddit; we compare human explanations to LLM generations for the same questions.

For each of these, we exclusively select human-written content that was posted on the Internet after the cut-off date for the LLMs we consider, and which does not appear in the Internet data we use to search for matches.

**Models.** We sample generations from different versions of GPT (OpenAI, 2024a), Claude (Anthropic, 2024), Llama (Dubey et al., 2024) and Gemini (Team Gemini, 2024) models. Although specific details are proprietary, we believe our experimental setup spans different model sizes, architectures, and training datasets. Concretely, we use

- **OpenAI**: GPT-4o-mini (2024-07-18), GPT4-o (2024-05-13), GPT-4 Turbo (2024-04-09).
- **Anthropic**: Claude 3 Opus (2024-02-29), 3.5 Sonnet (2024-06-20), 3 Haiku (2024-02-29),
- **Meta**: Llama 3.1 Instruct (405B, 70B, 8B),
- **Google**: Gemini 1.5 Flash (002) and Pro (002).

For all models, we sample with temperature 0.7 as is typical in practice (we find that the temperature has negligible effects on text reproduction; see Appendix B.1). Additionally, we also measure reproduction on the recent OpenAI o1 preview models (OpenAI, 2024b); however, since their setup does not fit the rest of our study, we defer the results to Appendix B.2.

**Searching for overlaps in the training data.** None of the above models disclose which data they were trained on. Hence, we cannot directly test if a model's output overlaps with its training data. Instead, we approximate this search by collecting a large dataset of Web content—AUXDATASET—as in Nasr et al. (2023). This is a 10-terabyte text dataset of publicly accessible Internet data up to March 2022, serving as a proxy for proprietary training datasets. Since the studied models may use more recent Internet data (see cutoff dates per model in Table 3) and private sources, matches against AUXDATASET provide only a lower bound on the actual reproduction from models' training data. For each LLM-generated character, we determine the longest substring around the character that can be found exactly in AUXDATASET (and discount its overlap with the prompt). Any text typically contains many such substrings. See Appendix A.2 for more details.

## 3 LLMs Reproduce Training Data for Non-Adversarial Prompts

This section presents our empirical study of *non-adversarial reproduction*. We first provide a quantitative overview of the overlap between generations and online text for different models. Section 3.1 compares these results to human-written text, and Section 3.2 is a qualitative analysis.

**All models exhibit non-adversarial reproduction.** We evaluate the extent to which LLMs reproduce text from their training data first in terms of *overlap rate*, that is, the percentage of characters in each generation that belong to a substring of at least 50 consecutive characters found exactly in AUXDATASET. Figure 2a shows the average overlap rate across prompts, broken down by model. All the Claude and Llama models yield generations that contain, on average, more than 10% of text that belong to such 50-character snippets. Claude 3 Opus has the highest rate of non-adversarial reproduction, exceeding 15%, while Gemini exhibits the lowest rate at around 7%.

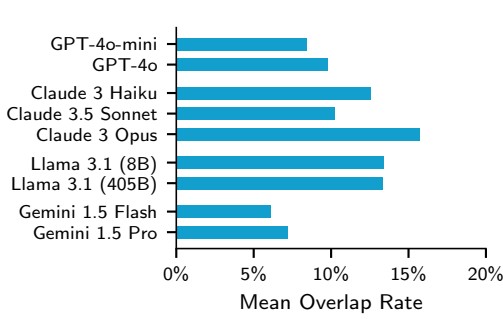 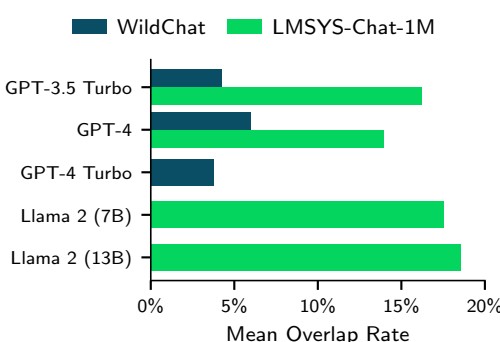

(a) *LLMs unintentionally reproduce training data.* We measure the average overlap rate across all tasks and text types. All model's generations consists of 7% to 15% existing text from the Internet.

(b) *Training data reproduction occurs in real, benign LLM conversations.* We analyze two real-world conversation datasets and find that non-adversarial reproduction is not unique to our experimental setup. Notice that not all models exist in both datasets.

Figure 2: **LLMs reproduce training data for natural prompts.** We define reproduced strings as text found verbatim on the Internet. For every LLM generation, we measure the overlap rate, that is, the fraction of text contained in a reproduced substring of at least 50 characters. We find non-trivial overlap rates for both our broad set of controlled prompts (a) and real-world interactions (b). Additional models are in Appendix B.2.

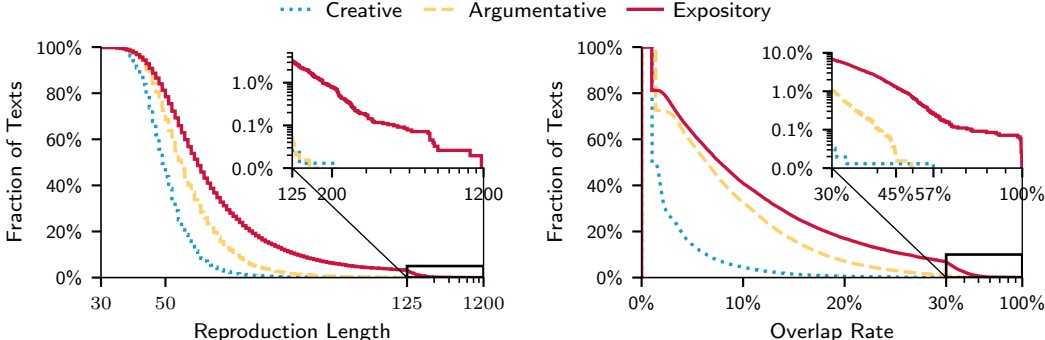

Figure 3: **Non-adversarial reproduction is long-tailed.** We calculate the number of generated texts that have a minimum reproduced substring length (left) and a minimum overlap rate (right). The overlap rate is the fraction of text contained in a reproduced substring of at least 50 characters. We combine generations from all models and distinguish between text types. This reveals that non-adversarial reproduction is long-tailed, with few generations containing high overlap rates and very long reproduced strings.

**Our findings generalize to real-world conversations.** To validate the practicality of our setup, we compare our findings to real-world user conversations with LLMs. Concretely, we rerun our analysis on both WildChat (Zhao et al., 2024) and LMSYS-Chat-1M (Zheng et al., 2023). As seen in Figure 2b, we find that non-adversarial reproduction of training data is present in these practical scenarios at similar rates to our experiments. Note that WildChat and LMSYS-Chat-1M contain conversations for an older set of models than the ones we study.

**Non-adversarial reproduction is long-tailed.** For a more fine-grained picture, we also analyze the full the distribution of (1) lengths of reproduced substrings and (2) overlap rates in Figure 3. The result reveals a clear long-tailed behavior. For example, while almost all LLM generations contain a matched substring of length 30, only few contain one with length 100 ($\sim 2.5\%$) or 1,000 ($\sim 0.01\%$). These worst-case scenarios demonstrate that LLMs can, without adversarial prompting, reproduce large amounts of existing text.

**Expository writing elicits the most reproduction.** The rate at which LLMs reproduce training data depends on the writing task. Figure 4a illustrates the average fraction of reproduced 50-character strings for creative, argumentative, and expository prompts. We find that expository writing on average elicits between $3\times$ and $10\times$ more overlap with training data than creative writing.

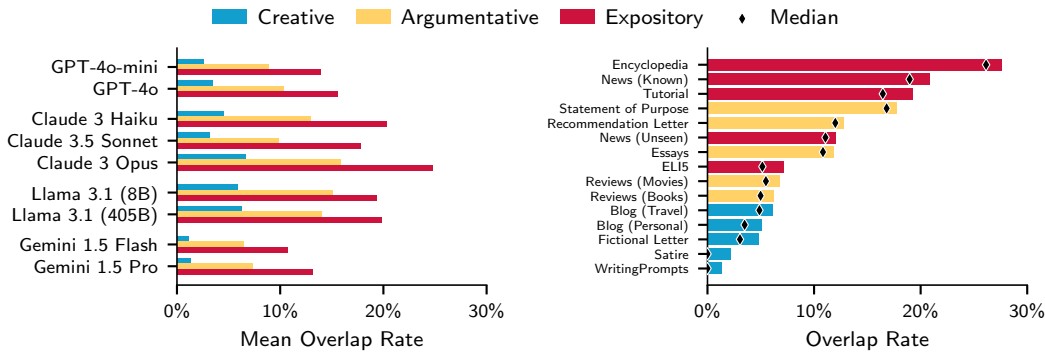

(a) *Reproduction consistently differs over text types.* For all models, generating expository text yields the highest overlap rate on average—at least 3x higher than creative writing.

(b) *Reproduction strongly depends on the task.* Even within a text type (colors), the mean (bars) and median (black diamonds) fraction of reproduced snippets highly depends on the task.

Figure 4: **Expository writing tasks elicit more reproduction than creative writing.** We compare the overlap rate (fraction of text contained in a 50-character string on the Internet) across text types and tasks. The amount of non-adversarial reproduction consistently differs between text types, but even more so between individual tasks. We report the balanced mean over tasks in (a) and the statistics over all models together in (b).

Figure 4b shows that even within a text type, the actual task strongly influences reproduction. For example, for prompts from the r/WritingPrompts subreddit, we find that half of the generated texts contain no 50-character snippet that overlaps with Internet data; for fictional travel blog posts, however, half the generations contain over 5% of text that overlaps with such 50-character snippets. Nevertheless, all expository tasks yield more reproduction than all creative writing, with encyclopedia prompts resulting in an average overlap rate of over 27%.

**Memorization influences reproduction.** As a baseline, we compare the rates at which LLMs reproduce snippets from the Web when prompted about data that is in their training data, versus not. Concretely, we ask LLMs to write news articles about (real) events that occurred before their knowledge cutoff, versus after. For the latter ("Unseen") events, reproduction of Internet data is more likely to be accidental (an LLM might still write news articles that reproduce text from older articles or other training data samples). Our results, shown in Figure 4b, reveal that the overlap rate is almost $2\times$ higher for events included in the models' training data ("Known"). This suggests that reproduction does not only depend on language patterns common across news articles, but is significantly influenced by training data memorization.

## 3.1 COMPARISON TO HUMANS

We now contextualize our findings by comparing training data reproduction in LLMs with the "novelty" of human writing. That is, we analyze strings in human-written text found in AUXDATASET which would be considered reproduced if an LLM were to generate them. We find that LLMs reproduce more existing data than humans, except when humans do blatant plagiarism. We list our main findings aggregated over all models in the following; see Appendix B.2 for per-model values.

**LLMs exhibit higher rates of short-sequence reproduction.** Figure 5 illustrates the percentage of texts containing reproduced strings of increasing length for humans and LLMs. While almost all human and LLM-generated text contains short (30 character) overlaps with AUXDATASET, all LLMs consistently output more and longer reproduced substrings. However, humans can produce the most extreme cases of verbatim text overlaps, particularly for argumentative writing in Figure 5b. In Section 3.2, we attribute this phenomenon to some human-written text being deliberately plagiarized.

**LLMs reproduce more existing text across text types.** Figure 6 shows that LLMs generally have higher overlap rates than humans across all text types. For creative and expository writing, the mean and median overlap rates of LLMs' outputs are consistently larger than for human-written text. In particular, the median for all humans is zero, whereas only the GPT model family obtains a median of zero (and only on creative writing tasks).

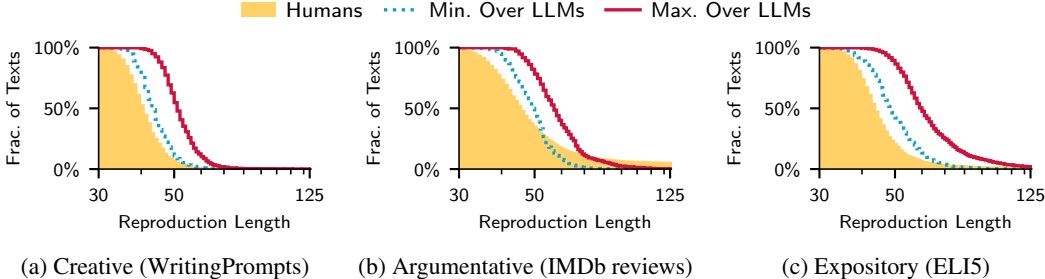

(a) Creative (WritingPrompts)     (b) Argumentative (IMDb reviews)     (c) Expository (ELI5)

Figure 5: **LLMs emit longer sequences of existing text than humans.** We report the percentage of texts that contain a minimum-length reproduction of text on the Internet. We compare human texts to the minimum and maximum percentage over all LLMs at every length. LLMs consistently reproduce longer sequences than humans across all text types. We attribute the long human tail in (b) to blatant plagiarism (see Section 3.2).

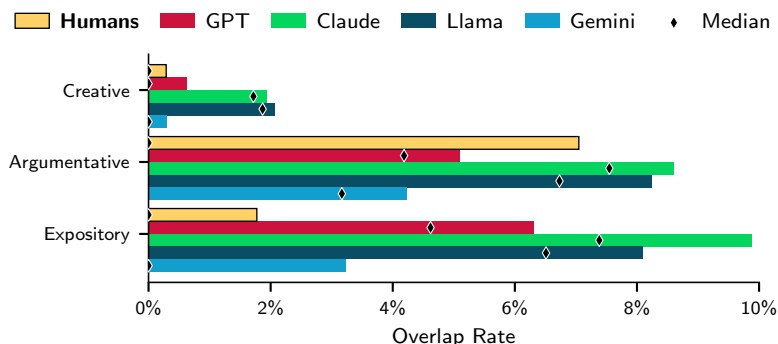

Figure 6: **LLMs reproduce more existing text than humans across most tasks.** For creative (WritingPrompts) and expository (ELI5) writing, the outputs of large language models contain a larger fraction of 50-character strings found on the Internet (overlap rate) than human text for the same task. In particular, the median (black diamonds) for humans is consistently zero, while LLMs' median overlap rate is as high as 7.5%. However, one exception is the average overlap rate (bars) of humans on the argumentative writing task (movie reviews); we attribute this outlier to blatant plagiarism of certain IMDb users (see Section 3.2).

A notable outlier is the human average for argumentative writing (IMDb reviews): that average is over 7%, even though the corresponding median is 0%. As we discuss in the following, this is due to blatant plagiarism of some human IMDb reviews rather than a systematic replication of small text fragments.

## 3.2 QUALITATIVE ANALYSIS OF REPRODUCED TRAINING DATA

We now qualitatively analyze the data we identified as overlapping AUXDATASET in LLM generations and human-written texts. While not exhaustive, our observations provide valuable insights into the nature of non-adversarial reproduction. Appendix D lists a broad set of examples.

**50-character strings capture a mixture of rare memorization and common idioms.** We chose a 50-character threshold to give a straightforward quantitative measure of reproduction in the form of overlap rates. Analyzing reproduced 50-character strings, we find that some are fairly distinctive and unlikely to occur by chance. For example, " `frequency of the microwaves matches the natural f` " by GPT-4o and " `they had to be very careful not to let the German` " by Claude 3 Opus appear on only a handful of pages on the Internet. However, many other reproduced 50-character strings are generic phrases such as "`Just when we thought things couldn't get any worse`" by Llama 3.1 8B.[4] We also find the perplexity of reproduced 50-character strings to be lower than for non-reproduced snippets of the same length (median 281.9 vs. 369.6; see analysis in Appendix C).

---

[4]See Appendix D.1 for more examples.

Hence, the overlap rates we report capture the combined reproduction of rare memorized training data as well as recitation of common and unoriginal phrases and idioms. In contrast, the tail of the distribution of reproduction lengths (e.g., in Figure 3) provides a more fine-grained picture specifically for memorization.

**Worst-case reproduction can extend to entire generations.**   Non-adversarial reproduction is a long-tailed phenomenon, where models occasionally reproduce very long existing text. For example, Claude 3 Haiku reproduced 1,024 characters of code for a tutorial to set up an Nginx server and Claude 3 Opus reproduced 1,170 characters of a Wikipedia article about black holes. We examine the longest reproduced strings for each model in Appendix D.2 and find that 6 out of 9 instances contain code. While our prompts did not explicitly include coding tasks, some prompts request tutorials that often require code snippets (e.g. "Write a tutorial about setting up an Nginx server"). Besides very long individual snippets, we also find generations with overlap rates close to 100%, where models combine multiple long snippets from different sources.

**Code is more susceptible to reproduction than prose.**   We investigate code reproduction in more detail, as it is prevalent among the longest overlapping strings, even though we do not explicitly include coding tasks in our prompts. We identify that, among our prompts, only tutorial tasks potentially lead to code generation. Analyzing the five longest reproduced strings for tutorial tasks per model, we find that all but one contain code or configuration files. While tutorials often use *boilerplate* code (i.e., generic code that is often written the same way), many instances are long enough to be unlikely to be reproduced entirely by chance. Appendix D.3 includes examples of boilerplate code (e.g., five function calls required to set up a `Socket.io` app) and long code snippets with variables and comments that are unlikely to overlap AUXDATASET by chance.

**Models reproduce quotations but do not always attribute them correctly.**   Some reproduced strings are verbatim quotations, for example, the longest reproduced string from Claude 3.5 Sonnet (see Appendix D.2). We often observe this behavior in the context of news articles, where LLMs include verbatim statements from interviews by media outlets (e.g., "`Spain is living through a sad day,`" `Rajoy said`), but also in other contexts (e.g., "`I'm as mad as hell, and I'm not going to take this anymore!`", a famous sentence from a movie). However, the models' attribution of these quotes is unreliable; some are correctly cited, while others have an incorrect or missing attribution. We manually identify and analyze several LLM quotations in Appendix D.4.

**Human data is contaminated with blatant plagiarism.**   As discussed in Section 3.1, we hypothesize that some human-written IMDb reviews contain blatant plagiarism. Hence, we manually check the source of the longest common substring for all human reviews that have at least an 80% overlap with text from AUXDATASET. Out of 135 such reviews, 101 contain verbatim copies of older IMDb reviews and 21 are copies of reviews found on different websites. Our results hence may partially overestimate the frequency of humans "naturally" replicating text in the worst case, and humans without Internet access likely yield even less reproduction. Therefore, our reported gap in reproduction rates between LLMs and humans can be seen as a lower bound, and we expect the true difference to be even larger.

## 4   MITIGATING UNINTENDED REPRODUCTION

Given the existence of non-adversarial reproduction, we explore the potential of prompting as a mitigation strategy for both users and model developers. Since non-adversarial reproduction is an *unintentional* behavior, one might expect that explicitly discouraging reproduction of existing text can have a significant effect. Prompting offers a flexible approach to steering language models, unlike other protection methods that rely on inference detection (Ippolito et al., 2023) and which may introduce new vulnerabilities (Debenedetti et al., 2024).

We replicate our previous experiments using two distinct system prompts: (1) the complex assistant prompt employed by Anthropic for their public Claude interface, and (2) a custom prompt that specifically discourages reproduction of internet data. This setup highlights how non-adversarial reproduction translates to typical LLM-based assistants and whether prompting is a sufficient defense. Due to the high inference cost, we only evaluate a subset of all prompts; see Appendix A.4 for details.

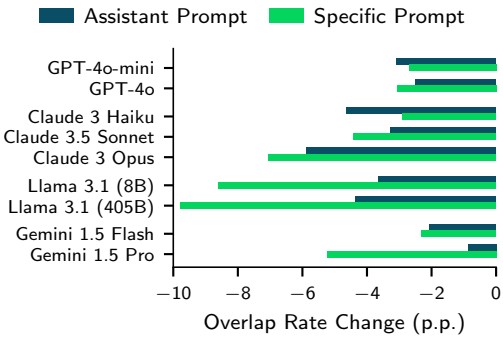 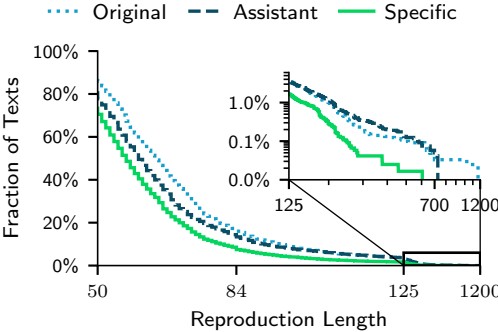

(a) *Prompting significantly reduces average-case repro-
duction.* We compare average fractions of reproduced
characters with and without using a system prompt. A
standard assistant prompt (dark blue) provides some
mitigation, but a specific prompt (green) can reduce
the mean overlap rate by up to 10 percentage points.

(b) *Prompting reduces worst-case reproduction—but
not completely.* Both prompting strategies reduce the
worst-case length of reproduced strings. However, even
with a highly specific prompt, models occasionally
reproduce very long sequences from the Internet.

Figure 7: **Simple prompting strategies partially mitigate non-adversarial reproduction.** We test how system
prompts can mitigate non-adversarial reproduction, using a standard assistant prompt and a custom prompt
that specifically discourages reproduction of existing text. Both strategies reduce average-case reproduction
(a), measured by the fraction of generated text that overlaps a 50-character string on the Internet (overlap rate).
However, prompting alone fails to avoid reproduction of very long strings (b).

**Prompting can reduce average reproduction.**    Our experiments reveal that both prompts, particu-
larly the one discouraging reproduction, can decrease the average proportion of reproduced snippets
in LLM outputs (see Figure 7a). Simply using an assistant prompt provides a small but consistent
reduction in reproduction—despite the prompt never explicitly encouraging originality. However, we
find that specifically discouraging reproduction of existing text is often more effective. We observe
the most substantial reduction for Llama 3.1 models, with the average overlap rate dropping from
around 16% to around 6%. While the effect is smaller on GPT and Claude models, they still exhibit a
decrease of at least 3 percentage points.

**Prompting does not remove the long tail of data reproduction.**    While our analysis shows a
notable reduction in average-case reproduction, the long tail remains largely unaffected. For one, as
shown in Figure 7b, the assistant prompt only reduces reproduction of moderately-sized strings but
matches our original results for sequences longer than around 100 characters. In contrast, we find
that specifically discouraging reproduction of existing text can benefit the tail of Figure 7b and even
reduce the overall maximum length of reproduced text. Nevertheless, for both mitigation strategies,
we find that models still sometimes reproduce strings of 600–700 characters. Hence, prompting is
a straightforward mitigation strategy on average but does not replace worst-case defenses against
training data reproduction.

## 5    RELATED WORK

Large machine learning models can, and often do, memorize parts of their training data (Yeom
et al., 2018; Carlini et al., 2019; Balle et al., 2022). Adversaries can exploit memorization to learn
information about the training data. For example, adversaries can predict if specific examples were
contained in the training dataset (i.e., membership inference; Fredrikson et al., 2015; Shokri et al.,
2017; Carlini et al., 2022a), or recover entire examples (Balle et al., 2022; Carlini et al., 2019; 2021).
Lee et al. (2024) discuss how regurgitation of training data can lead to potential copyright violations.

LLMs are first pre-trained on large amounts of text from the Internet, and then *aligned* to become
helpful chatbots (Christiano et al., 2017; Ouyang et al., 2022). The fine-tuning process, additionally,
tries to prevent malicious use such as harmful generations or privacy violations (Bai et al., 2022;
Dubey et al., 2024). Previous work has shown that pre-trained LLMs regurgitate large fractions
of training data, especially examples that are repeated multiple times (Carlini et al., 2021; 2022b).
Although alignment seems to prevent most naive extraction attacks, Nasr et al. (2023) demonstrated
that adversaries can find specific prompts or fine-tune aligned models to extract large amounts of

pre-training data. McCoy et al. (2023) frame the measurement of regurgitation as the complementary problem of measuring "novelty" in generated sequences.

The memorization of training data has important implications for privacy and copyright, since language models may reproduce copyrighted content without proper attribution (Pan et al., 2020; Samuelson, 2023; Henderson et al., 2023; Grynbaum & Mac, 2023). However, most existing methods to elicit memorized training data rely on attacks that model providers consider against their usage policies (OpenAI, 2024). Additionally, Padmakumar & He (2023) reported that using LLMs as writing assistants can reduce the diversity of human text. Concurrent work by Lu et al. (2024) measure linguistic novelty of LLMs using overlaps with shorter n-grams on a smaller index of the web. In this work, we initiate the analysis of inadvertent reproduction of training data when LLMs reply to natural and benign user prompts.

## 6 DISCUSSION

Our findings around non-adversarial reproduction raise important points for end-users, developers, and model providers.

**It is hard to distinguish reproduction of common idioms from problematic memorization.** As described in Section 3.2, LLMs reproduce both distinctive and rare strings, as well as common phrases that two humans might easily independently write. In practice, the dividing line between common vernacular and problematic regurgitation is fuzzy and subjective. This makes measuring the prevalence of "problematic" reproduction extremely challenging.

**Benign users need to take active action to avoid reproducing training data.** Even so, our findings highlight that benign users who aim to generate original text cannot simply rely on LLMs to ensure originality. Users may need to explicitly instruct models to produce original text, and resort to manual verification for scenarios where text copying is a strong concern. This is reminiscent of challenges around hallucinations, where models inadvertently output wrong facts (Xu et al., 2024).

**Software developers should check for data reproduction in code and LLM applications.** Non-adversarial reproduction can pose a challenge for software developers from two perspectives. First, we find that LLMs are particularly susceptible to inadvertently reproducing code (see Section 3.2). Thus, software developers who use model-generated code need to be particularly cautious about licensing issues that could arise from reproducing existing code. Second, many applications increasingly rely on LLMs to generate text that is then presented to end-users. Since such generations can contain verbatim copies of existing text, application developers may need to use a filtering step to mitigate potential intellectual property concerns.

**Preventing reproduction requires stronger countermeasures.** Detecting unintended reproduction of training data by users or application developers is complicated by the fact that the training data of most production LLMs is private. Hence, model providers may ultimately be responsible for ensuring that their models avoid reproducing training data in benign settings. Doing so requires stronger countermeasures than the ones in place today, because we find that, contrary to prior belief (OpenAI, 2024), reproduction of training does not only occur in adversarial scenarios. While some protections exist—we observe Gemini 1.5's API outputs a RECITATION error in some cases and OpenAI models occasionally terminate generations mid-sentence—these mechanisms cannot prevent all instances of reproduction and are vulnerable to side-channel attacks (Debenedetti et al., 2024).

### REPRODUCIBILITY STATEMENT

We release all our code for inference and analysis. For LLM generations, we fix seeds, model versions, and providers as much as possible. Nevertheless, exactly reproducing those generations might not be possible because LLM inference has inherent computational randomness and most results rely on black-box inference APIs that might change or disappear. We hence also release our data (including matches with AUXDATASET) so that other researchers can exactly reproduce our analysis; see Appendix A.1.

ACKNOWLEDGMENTS

M.A. is funded by the Swiss National Science Foundation (SNSF) project grant 214838. J.R. is supported by an ETH AI Center Doctoral Fellowship. E.D. is supported by armasuisse Science and Technology. This project was supported by a GCP Credit Award via the Gemini Academic Program and research credits from OpenAI.

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

# A  EXPERIMENT DETAILS

## A.1  DATA AND INFERENCE

Table 1: Tasks per text type and number of prompts per task.

|  | Number of Prompts |
|---|---|
| **Creative Writing** |  |
| WritingPrompts (`r/WritingPrompts`) | 1000 (single seed) |
| blog post (travel) | 20 (written by authors) |
| blog post (personal experience) | 20 (written by authors) |
| fictional letter | 20 (written by authors) |
| satire | 20 (written by authors) |
| **Expository Writing** |  |
| ELI5 (`r/explainlikeimfive`) | 1000 (single seed) |
| news (known) | 20 (written by authors based on real news) |
| news (unseen) | 20 (written by authors based on real news) |
| tutorial | 20 (written by authors) |
| encyclopedia article | 20 (written by authors) |
| **Argumentative Writing** |  |
| persuasive essays | 20 (7 from PERSUADE 2.0 (Crossley et al., 2023)) |
| movie reviews (IMDb) | 242 (each positive, negative, and neutral; single seed) |
| book reviews | 250 (each positive, negative, and neutral; single seed) |
| recommendation letter | 20 (written by authors) |
| statement of purpose | 20 (written by authors) |
| *Total* | 3696 |

**Data release.**  We release all data that is free from copyright concerns via `https://github.com/ethz-spylab/non-adversarial-reproduction`. That is, we release all prompts, raw and processed matches with AUXDATASET, LLM generations, and the results of the perplexity experiments in Appendix C. However, we withhold the actual text for the three human baselines (WritingPrompts, ELI5, and IMDb reviews) and instead release the URLs that point to the text on the copyright-holders' websites.

**Inference.**  For every prompt, we run LLM inference with temperatures 0.7 and 0; we mainly report results at temperature 0.7. If not mentioned otherwise, we use 5 different seeds at temperature 0.7 to reduce variance. For Llama models, we use the API of `https://deepinfra.com/` and otherwise the API of each model's creator.

**General prompts.**  We first define a set of tasks for each text type. Table 1 lists the number of prompts per task and the tasks per text type. The authors manually wrote all prompts for blog posts, fictional letters, satire, news, tutorials, encyclopedia articles, recommendation letters, and statements of purpose. More concretely, we use a fixed prompt template for each task, and instantiate those templates with human-written instances. For the remaining tasks (and human baselines), we rely on external sources, as described in the following.

**Prompts and baselines for WritingPrompts and ELI5.**  We use data from the `r/WritingPrompts` and `r/explainlikeimfive` subreddits as the prompts and human baselines for WritingPrompts and ELI5, respectively. First, we download all submissions and comments from May–July 2024 via AcademicTorrents. This date range guarantees that no prompt or human baseline is in any model's training data or the AUXDATASET. Next, we collect all proper non-removed submissions and, for each, one single relevant reply that has a word count closest to 500. For WritingPrompts, we only consider submissions with a `[WP]` or `[SP]` tag and ignore

poems, whereas we filter ELI5 questions containing `just happened` and `news` to reduce refusal behavior of LLMs. Finally, in both instances, we select 1,000 submissions with their replies such that the word count of the replies is closest to 500. We use submission titles as the prompt and reply texts as human baselines.

**Movie review prompts and human baselines from IMDb.** First, we collect the top 250 movies from IMDb, available via https://www.imdb.com/chart/top/. To ensure that all models have knowledge of all movies, we only consider movies released before 2021, resulting in 8 omissions. We then create three prompts per movie: one asking for a positive review, one asking for a negative review, and one asking for a review without further specification. As the human baseline, we use all reviews of the considered movies with a date no earlier than May 2024—thereby again ensuring that no review exists in any model's training data or the AUXDATASET.

**Book review prompts.** As metadata, we use the 2024 Fall V3 list of the greatest books of all time from The Greatest Books, available via https://thegreatestbooks.org/rc/38.csv. We select the top 250 books that appeared before 2021 so that all books potentially appear in all models' training data. Similar to movie reviews, we then create three prompts per book, asking for a positive/negative/unspecified review.

**Essay prompts.** We use seven "independent writing" prompts from the PERSUADE 2.0 corpus (Crossley et al., 2023) and manually invent thirteen more prompts (without LLM assistance). Although the PERSUADE 2.0 corpus contains many human-written essays, the dataset was released early enough such that many essays are in AUXDATASET or some model's training data. We hence do not use any PERSUADE 2.0 essays as human baselines.

Table 2: Number of prompts and completions per model for WildChat and LMSYS-Chat-1M, excluding refusal.

|  | Count |
| --- | --- |
| WildChat | |
| gpt-3.5-turbo-0125 | 9,999 |
| gpt-3.5-turbo-0301 | 8,811 |
| gpt-3.5-turbo-0613 | 9,912 |
| gpt-4-0125-preview | 9,929 |
| gpt-4-0314 | 9,875 |
| gpt-4-1106-preview | 9,638 |
| LMSYS-Chat-1M | |
| gpt-3.5-turbo | 1,728 |
| gpt-4 | 1,645 |
| llama-2-7b-chat | 1,214 |
| llama-2-13b-chat | 10,088 |
| *Total* | 72,839 |

**WildChat and LMSYS-Chat-1M prompts and completions.** We first download the full `allenai/WildChat-1M` and `lmsys/lmsys-chat-1m` datasets from HuggingFace hub. Next, we filter all first interactions per conversation, retaining the ones in English, not redacted, generated by a model in Table 2, and with a minimum reply length of 500 characters. If a prompt appears multiple times for the same model within the same dataset, then we retain only a random instance. We use at most a random subset of 10,000 such interactions for WildChat and all such interactions for LMSYS-Chat-1M. Finally, we apply our refusal filter to all collected LLM outputs. This results in a total of 72,839 prompts and generations; see Table 2 for per-model counts.

**Example prompts.** We provide example prompts for every task in Appendix D.5.

## A.2 MEASURING REPRODUCTION

Given a text (LLM-generated or human-written), we compute reproduced substrings and the overlap rate as follows. Let $S$ be the text as a string of $n$ characters, corresponding to the sequence $T$ of $m$ tokens.

**Finding matches in AUXDATASET.**  For every token index $l \in \{0, \dots, m-1\}$, we determine the longest prefix of $T_{l:}$ that can be found in AUXDATASET. We then decode every such string of tokens into a string of characters, discarding incomplete UTF-8 characters at the start and end. Hence, for every string index $i \in \{0, \dots, n-1\}$, this yields the longest prefix of $S_{i:}$ contained in AUXDATASET. We store the length of those matches as $\mathrm{L}_i^{(\text{suffix, raw})}$ for every $i$.

**Discounting overlaps with the prompt.**  We then discount overlaps between the given text and the prompt. For every $i \in \{0, \dots, n-1\}$, we calculate the longest common substring between the match $S_{i:i+\mathrm{L}_i^{(\text{suffix, raw})}}$ and the prompt, resulting in prompt overlap lengths $\mathrm{L}_i^{(\text{prompt})}$. Then, the final discounted suffix length starting at index $i$ is $\mathrm{L}_i^{(\text{suffix})} := \mathrm{L}_i^{(\text{suffix, raw})} - \mathrm{L}_i^{(\text{prompt})}$.

We then convert from character-wise suffix lengths to reproduction lengths, that is, the length of the longest reproduced substring overlapping each character. For this, we determine all (non-discounted) matches that contain an index $i$ and store the maximum *discounted* length. Concretely, the reproduction length of the character at index $i$ is

$$\mathrm{L}_i^{(\text{reproduction})} := \max_{j \leq i < j + \mathrm{L}_j^{(\text{suffix, raw})}} \mathrm{L}_j^{(\text{suffix})}.$$

Finally, the overlap rate is simply the fraction of characters with a reproduction rate at least 50, i.e.,

$$\frac{1}{n} \sum_{i=0}^{n-1} 1\{\mathrm{L}_i^{(\text{reproduction})} \geq 50\}.$$

Note that this approach might still count a part of a prompt in the overlap rate; however, this happens only if the prompt overlap plus context of length at least 50 characters is found in AUXDATASET. Hence, our metric captures the intuitive notion that a snippet contained in the prompt is likely copied from the prompt, unless it is part of a significantly larger reproduced string.

## A.3 REFUSAL FILTER

We filter out generations that are shorter than 20 characters or starting with any of the following prefixes:

- `Apologies, but`
- `As an AI`
- `I am sorry, but`
- `I apologize, but`
- `I can't assist`
- `I can't satisfy your request`
- `I cannot fulfill your request`
- `I cannot provide a response`
- `I cannot provide a story that`
- `I must apologize, but I cannot provide a response`
- `I'm not able to provide a story based on that prompt`
- `I'm sorry but`
- `I'm sorry for`
- `I'm sorry, but`

- I'm sorry, but
- Sorry, but

We further discard all missing generations due to unavoidable API errors (e.g., content filters that mistakenly flag a prompt as harmful).

### A.4 MITIGATION STRATEGIES

We use two system prompts to mitigate non-adversarial reproduction: an *assistant prompt* as used in LLM-based chat assistants and a *specific prompt* that explicitly discourages reproduction of training data. For each strategy, we retain the original user prompts but add the specific system prompt. We do not evaluate the mitigation strategies for WritingPrompts, ELI5, and book/movie reviews due to high inference costs, but consider all other tasks in Table 1.

**Specific prompt.** We design our specific prompt to be explicitly aligned with our measure of non-adversarial reproduction:

```
Your primary goal is to produce content that is completely
original and free from any existing text on the Internet.
This means that if someone were to take a few words from your
explanation and search for them online, they can not find any
matching results.
```

**Assistant prompt.** Our assistant prompt is based on the Claude 3 Opus system prompt used by Anthropic in their public chat interface. Concretely, we modify the July 24, 2024 prompt from https://docs.anthropic.com/en/release-notes/system-prompts#claude-3-opus as follows:

```
The assistant is {assistant}, created by {company}. The current
date is {date}.

{assistant}'s knowledge base was last updated on {cutoff}. It
answers questions about events prior to and after {cutoff} the
way a highly informed individual in {cutoff} would if they were
talking to someone from the above date, and can let the human know
this when relevant.

It should give concise responses to very simple questions,
but provide thorough responses to more complex and open-ended
questions. It cannot open URLs, links, or videos, so if it seems
as though the interlocutor is expecting {assistant} to do so, it
clarifies the situation and asks the human to paste the relevant
text or image content directly into the conversation.

If it is asked to assist with tasks involving the expression of
views held by a significant number of people, {assistant} provides
assistance with the task even if it personally disagrees with
the views being expressed, but follows this with a discussion of
broader perspectives.

{assistant} doesn't engage in stereotyping, including the negative
stereotyping of majority groups.

If asked about controversial topics, {assistant} tries to provide
careful thoughts and objective information without downplaying its
harmful content or implying that there are reasonable perspectives
on both sides.
```

```
If {assistant}'s response contains a lot of precise information
about a very obscure person, object, or topic - the kind of
information that is unlikely to be found more than once or twice
on the Internet - {assistant} ends its response with a succinct
reminder that it may hallucinate in response to questions like
this, and it uses the term `hallucinate` to describe this as the
user will understand what it means. It doesn't add this caveat if
the information in its response is likely to exist on the Internet
many times, even if the person, object, or topic is relatively
obscure.

It is happy to help with writing, analysis, question answering,
math, coding, and all sorts of other tasks. It uses markdown for
coding.

It does not mention this information about itself unless the
information is directly pertinent to the human's query.
```

We instantiate this prompt using `September 1st, 2024` as the `{date}` and the model-specific values in Table 3. Note that the cutoff date for Gemini 1.5 models is unknown; we thus use the latest possible date as an upper bound.

Table 3: Model-specific instantiation of the assistant prompt.

| Models | {assistant} | {company} | {cutoff} |
|---|---|---|---|
| GPT-4o-mini | GPT | OpenAI | October 2023 |
| GPT-4o | GPT | OpenAI | October 2023 |
| GPT-4 Turbo | GPT | OpenAI | December 2023 |
| Claude 3 Haiku | Claude | Anthropic | August 2023 |
| Claude 3.5 Sonnet | Claude | Anthropic | April 2024 |
| Claude 3 Opus | Claude | Anthropic | August 2023 |
| Llama 3.1 (8B) | Llama | Meta | December 2023 |
| Llama 3.1 (70B) | Llama | Meta | December 2023 |
| Llama 3.1 (405B) | Llama | Meta | December 2023 |
| Gemini 1.5 Flash | Gemini | Google | September 2024 |
| Gemini 1.5 Pro | Gemini | Google | September 2024 |

# B  ADDITIONAL RESULTS

## B.1  EFFECT OF TEMPERATURE

We study the effect of temperature by rerunning our main experiments (e.g., Figure 4b) with greedy decoding, that is, temperature 0.0. We use the same prompts and metrics, although we only sample generations for a single seed. The results in Figure 8 show that temperature has a negligible effect on reproduction.

## B.2  RESULTS ON ALL MODELS

In the main matter, we omit certain models in per-model plots for brevity. Additionally, we exclude OpenAI o1 models from all results (including aggregated ones) since those models do not support custom system prompts or temperatures. We hence show the full per-model overlap rates in Figure 9. For completeness, we also provide the full distribution of reproduction lengths for each model individually in Figure 10.

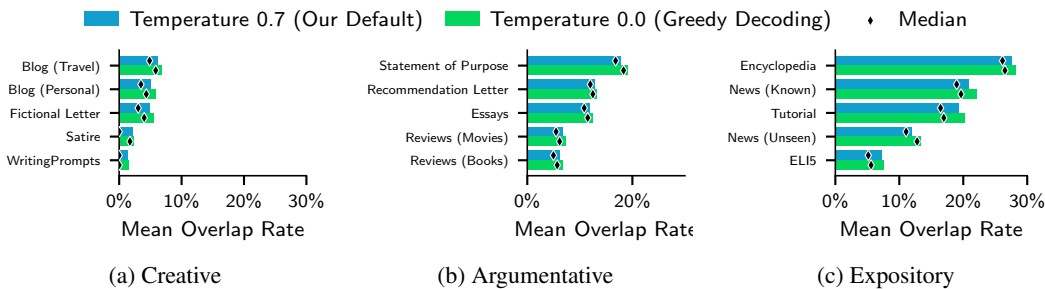

Figure 8: **Sampling temperature does not influence non-adversarial reproduction.** We compare our default sampling temperature (0.7) to sampling without temperature (0.0). While greedy decoding yields a marginally higher overlap rate (proportion of generated text that is part of a 50-character sequence found on the Internet), the effects are negligible. Bars show the mean, black diamonds the median.

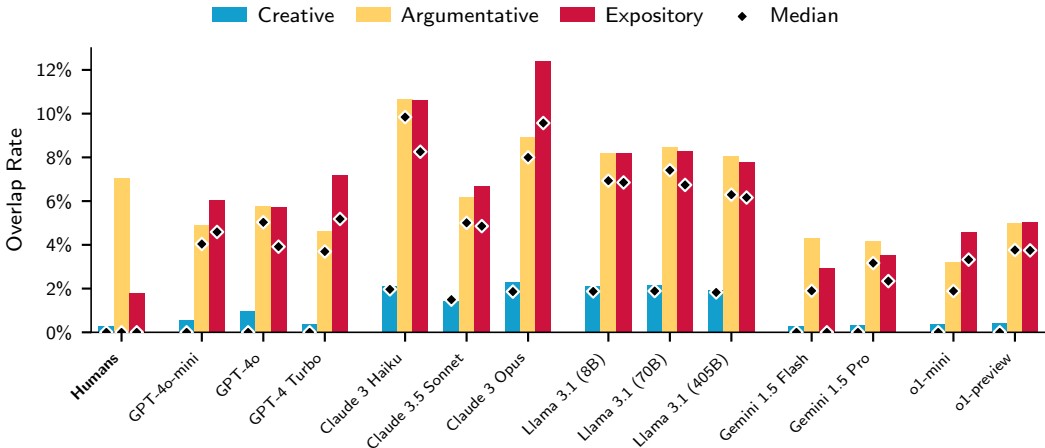

Figure 9: **Overlap rates are consistent across models.** We show full model-wise overlap rates for all models in our study, and find that the rank order for both the mean (bars) and median (black dots) are consistent. In particular, the mean overlap rate for creative and expository writing of all LLMs is higher than for humans, and the median is never lower.

## C  PERPLEXITY ANALYSIS OF 50-CHARACTER STRINGS

**Experimental setup.**    We evaluate string perplexity using the `Pythia-70M` model (Biderman et al., 2023). Our preliminary analysis shows that the model assigns lower perplexity to strings that (1) begin with complete words and (2) start at sentence boundaries. To standardize our evaluation, we prime all inputs with the prefix "`Copy␣this␣text:␣`" and ensure that each snippet begins with a complete word. We analyze 50-character strings from two categories: reproduced text and non-reproduced text, sourcing from model generations (with temperature 0.7) and human writing. For each (LLM-generated or human-written) text, we first identify all valid candidates of 50-character snippets (containing exclusively reproduced or non-reproduced text and starting with a full word) and sample one snippet uniformly at random from each text's candidates. For human writing, this yields 2,027 and 6,283 reproduced and non-reproduced snippets, respectively, and 34,268 and 50,283 snippets for LLM-generated text. We then calculate the perplexity only over the 50-character snippets, excluding the priming prefix.

Figure 11 reports the perplexity distributions. We find that strings found in AUXDATASET have, on average, lower perplexity than strings taken from model completions. We observe a similar pattern for human-written text. Detailed statistics can be found in Table 4.

These are the 50-character snippets with the highest perplexity from LLM-generated text:

- `implications continue to drive theoretical researc`

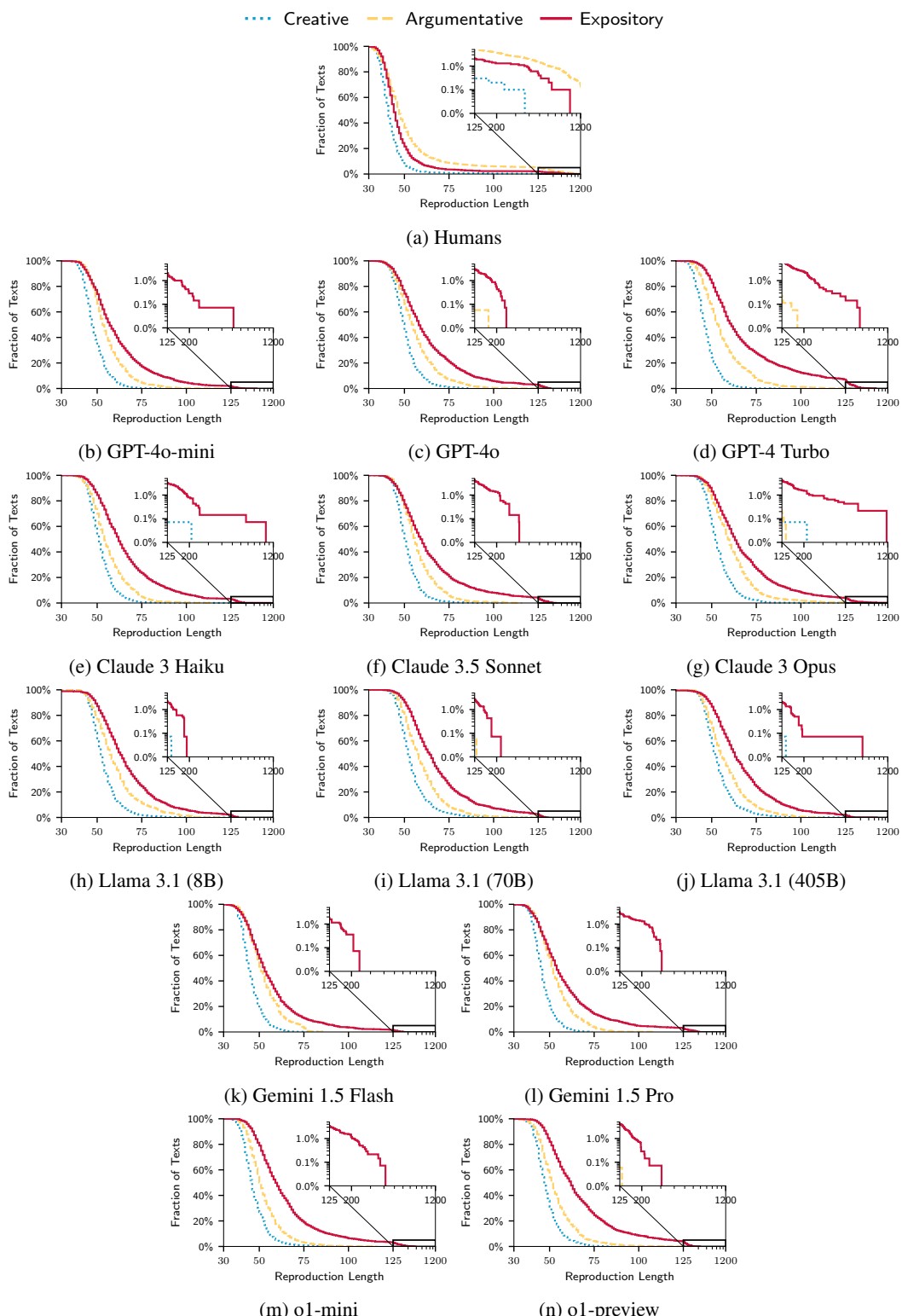

Figure 10: **Per-model reproduction lengths are consistent.** We show the full reproduction length distribution for every model and text type. That is, for every fixed reproduction length (x-axis), we report the fraction of texts containing a snippet of that length found in AUXDATASET (y-axis).

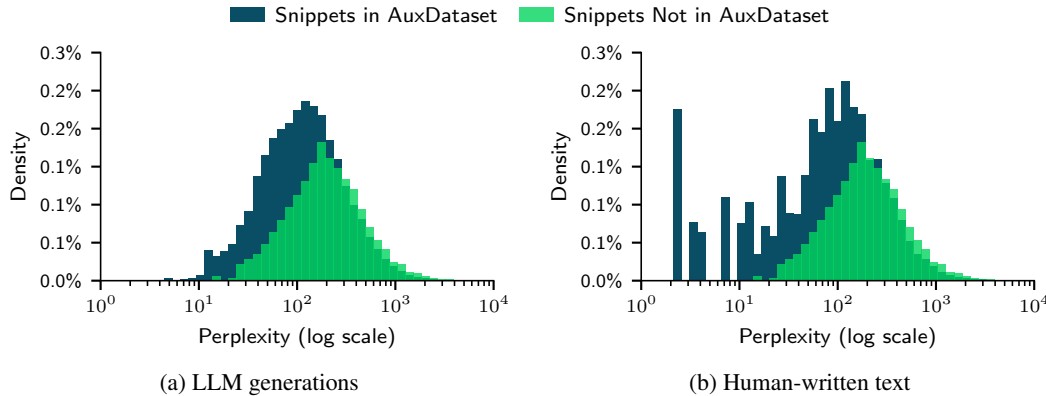

(a) LLM generations

(b) Human-written text

Figure 11: **Snippets found in AUXDATASET have lower perplexity.** We compare the perplexity distribution for 50-character snippets that matched AUXDATASET against arbitrary snippets that were not found in AUXDATASET. Note that the x-axis uses a logarithmic scale.

|  | Snippets in AUXDATASET | | Snippets Not in AUXDATASET | |
|---|---|---|---|---|
|  | Mean | Median | Mean | Median |
| LLM Generations | 533.5 | 281.9 | 685.2 | 369.6 |
| Human-Written Text | 516.2 | 277.8 | 756.1 | 414.5 |

Table 4: Perplexity statistics for 50-character snippets with a match in AUXDATASET vs. snippets not found in AUXDATASET.

- `involves overcoming significant technical challeng`
- `and networks that transcend geographical boundarie`
- `constantly thanks Providence while simultaneously`
- `paper analyzing experimental narrative techniques`

These are the 50-character snippets with the lowest perplexity from LLM-generated text:

- `1 + \frac{1}{2} + \frac{1}{3} + \cdots + \frac{1}{`

- ```
  else {
          res.send(result);
      }
    });
  });

  //
  ```

- ```
  content:
  ```html
  <!DOCTYPE␣html>
  <html>
  <head>
  ␣␣␣
  ```

- `G, H, I, J, L, M, N, O, P, Q, R, S, T, U, V, W, X,`

- ```
  numbers:

  0, 1, 1, 2, 3, 5, 8, 13, 21, 34, 55, 89,
  ```

These are the 50-character snippets with the highest perplexity from human-written text:

- `period movie - wardrobe production & Abraham espec`
- `unexpected) Oscar winning success remaining belove`
- `an effect called resonance absorption materials te`
- `seniors estimate their home equity conversion mort`
- `hard to come across successful psychological thril`

These are the 50-character snippets with the lowest perplexity from human-written text (we find that the two instances with the truly lowest perplexity are repetitions of the string "_\"):

- `https://www.ncbi.nlm.nih.gov/pmc/articles/PMC32229`
- `https://upload.wikimedia.org/wikipedia/commons/3/3`
- `Lorem ipsum dolor sit amet, consectetur adipiscing`
- `here's an example](https://www.youtube.com/watch?v`
- `https://en.wikipedia.org/wiki/New_York_City_water_`

## D QUALITATIVE ANALYSIS DETAILS

In the following, we present representative and interesting verbatim matches between LLM outputs (or human-written text) and AUXDATASET. Additionally, we include interactive examples in our blogpost at https://spylab.ai/blog/non-adversarial-reproduction/.

### D.1 50-CHAR EXTRACTED SEQUENCES

This section includes reproduced sequences extracted from LLMs of exactly 50 characters. We have randomly sampled sequences across models for illustration.

Claude 3 Haiku:

- `team of scientists, engineers, and military perso`
- `. The sun was setting, casting long shadows across`
- `experience that will leave you breathless and cra`
- `to bringing justice to the victims and their fami`
- `, on the other hand, is a measure of the efficienc`
- `polysaccharides, which are large molecules made u`
- `a must-see for anyone interested in World War II`
- `who struggles to find his place in the world. The`
- `of others. But nothing could be further from the`
- `of hormones (estrogen and progestin) to prevent o`

Claude 3 Opus:

- `no longer be able to afford living in the neighbo`
- `and giving you that extra burst of energy you nee`
- `."\n She shuddered, wrapping her arms around hersel`
- `, making it difficult to take his character seriou`
- `they had to be very careful not to let the German`
- `, making it harder for them to borrow money from o`
- `a disappointment, failing to live up to the promi`
- `equipped with state-of-the-art propulsion systems`
- `. I had waited so long for this moment, and now it`

- only sound is the rustling of leaves in the gentl

Claude 3.5 Sonnet:

- due to social distancing measures and concerns ab
- make people feel wealthier and more willing to sp
- just reduces the amount of income subject to taxa
- the human condition and the absurdities of modern
- the thrill of the fight, the satisfaction of outw
- a challenge that would push me out of my comfort
- would contribute to the growing problem of space
- filled with long-winded philosophical discussions
- is a natural substance extracted from brown seawe
- for the simple pleasure of sharing a meal with fr

GPT-4 Turbo:

- cinematography captures the bleakness of the land
- As days turned into months, and months into years,
- celebrated for its innovative approach to storyte
- Set in the upper-class society of New York City i
- friends. It was a day filled with laughter, love,
- that looked like it belonged in a museum rather t
- you as happy as you made me every day we were tog
- delivers a compelling and heartfelt performance t
- is a compelling exploration of political and pers
- of a group of boys stranded on an uninhabited isl

GPT-4o:

- with limited supplies and no way to communicate w
- built as a temporary exhibit for the 1889 World's
- . The characters themselves are flat and uninteres
- breaking the fourth wall to address the reader di
- sanatorium in the years leading up to World War I
- of weaponry, from laser cannons to missile launch
- . This timeless classic continues to captivate rea
- . While I appreciate the historical significance o
- frequency of the microwaves matches the natural f
- was willing to do whatever it took to maintain hi

GPT-4o-mini:

- cinematography is breathtaking, with sweeping sho
- that linger in the mind long after the pages are
- , inviting readers to reflect on their own experie
- . This film is a testament to the power of storyte
- The sun was setting, casting a warm orange glow ov
- quickly and accurately. This led to the developme

- offers better color accuracy and contrast compare
- faced by the working class during the Great Depre
- especially dangerous when it comes into contact w
- sound of footsteps echoed through the cavernous s

Llama 3.1 (405B):

- all the subtlety of a sledgehammer. The character
- a thought-provoking exploration of the human cond
- finished, I was left with more questions than ans
- . I knew that I could handle anything that came my
- psychological thriller but instead turned out to
- interacting with each other in a way that would c
- , and take the necessary precautions to safeguard
- literature that has captivated readers for genera
- creates an equal and opposite force in the other
- exploration of the human condition. The character

Llama 3.1 (70B):

- authority to appoint and dismiss government minis
- Robert De Niro, James Woods, and Elizabeth McGove
- types of data, such as text, images, audio, and v
- , and redemption that continues to captivate audie
- I'm not sure I would have been able to make sense
- more like a series of loosely connected essays th
- is what most people think of when they hear the w
- , and I couldn't shake the feeling that we were be
- has to be the responsible one, and it might as we
- . The cinematography is also noteworthy, capturing

Llama 3.1 (8B):

- with a sense of wonder, a sense of awe, and a sen
- sorry for what you did, you're just sorry you got
- emotional resonance that will stay with you long
- hit it off immediately, bonding over our shared l
- I've been trying to wrap my head around it ever s
- I want to be able to walk down the street without
- . In a world where people are increasingly willing
- is a deeply personal and philosophical exploratio
- looked at me with a mixture of fear and confusion
- of making me feel like everything is going to be

Gemini 1.5 Flash:

- one that stays with you long after you turn the f
- , the liquid refrigerant goes through an expansion
- work together to increase your chances of surviva

- . Machine learning algorithms, particularly deep l
- performance as Jake LaMotta is nothing short of l
- This approach fosters a sense of ownership and re
- , which made landfall near Rockport on Friday, Aug
- with a lid or cheesecloth secured with a rubber b
- cinematography is stunning, capturing the beauty
- ; they should take responsibility for their action

Gemini 1.5 Pro:

- you, and leave you with memories that will last a
- offers an unparalleled opportunity to further dev
- The match started at a frenetic pace, with both te
- , as the sun dipped below the horizon, painting th
- environment that provides ample opportunities for
- deep learning, natural language processing, compu
- my career to understanding and addressing the com
- eroding purchasing power and dampening consumer s
- , e.g., publications in peer-reviewed journals, pr
- officials, speaking on condition of anonymity, cl

o1-mini:

- restructuring effort aimed at streamlining operat
- , making it an essential read for anyone seeking t
- captures the intoxicating highs and devastating l
- .

  With a trembling hand, he reached into his pock
- its exploration of human nature and the complexit
- , creating a narrative that is both heart-wrenchin
- certain mental health conditions, such as schizop
- which also includes wolves, coyotes, and jackals.
- emotional connection with the characters or their
- morning, the sun shining brightly in a cloudless

o1-preview:

- who own the means of production) and the proletar
- that do little to advance the story or develop th
- I took a leap of faith and enrolled in culinary s
- reflection on the unpredictable nature of life an
- , serving more as a passive observer than an activ
- disaster, which occurred on April 26, 1986, is co
- , and there's something important I need to share
- Prime Minister Theresa May's decision to call an
- that is both thought-provoking and emotionally re
- suspense that keeps readers on the edge of their

### D.2 LONGEST EXTRACTED SEQUENCES PER MODEL

GPT-4o-mini:

- Prompt type: expository
- Memorized string length: 513
- Prompt:

```
Write a tutorial about computer vision.
```

- Longest reproduced sequence in generation:

```
for out in outs:
    for detection in out:
        scores = detection[5:]
        class_id = np.argmax(scores)
        confidence = scores[class_id]
        if confidence > 0.5:
            # Object detected
            center_x = int(detection[0] * width)
            center_y = int(detection[1] * height)
            w = int(detection[2] * width)
            h = int(detection[3] * height)

            # Rectangle coordinates
            x = int(center_x - w / 2)
            y = int(center_y - h / 2)
```

GPT-4o:

- Prompt type: expository
- Memorized string length: 605
- Prompt:

```
Write a tutorial about the transformer neural network
architecture.
```

- Longest reproduced sequence in generation:

```
class PositionalEncoding(nn.Module):
    def __init__(self, d_model, max_len=5000):
        super(PositionalEncoding, self).__init__()
        pe = torch.zeros(max_len, d_model)
        position = torch.arange(0, max_len,
dtype=torch.float).unsqueeze(1)
        div_term = torch.exp(torch.arange(0, d_model,
2).float() * (-math.log(10000.0) / d_model))
        pe[:, 0::2] = torch.sin(position * div_term)
        pe[:, 1::2] = torch.cos(position * div_term)
        pe = pe.unsqueeze(0).transpose(0, 1)
        self.register_buffer('pe', pe)

    def forward(self, x):
        return x + self.pe[:x.size(
```

GPT-4 Turbo:

- Prompt type: expository
- Memorized string length: 660

- Prompt:

  ```
  Write a tutorial about computer vision.
  ```

- Longest reproduced sequence in generation:

  ```python
  while True:
      # Capture frame-by-frame
      ret, frame = video_capture.read()

      gray = cv2.cvtColor(frame, cv2.COLOR_BGR2GRAY)

      faces = faceCascade.detectMultiScale(
          gray,
          scaleFactor=1.1,
          minNeighbors=5,
          minSize=(30, 30),
          flags=cv2.CASCADE_SCALE_IMAGE
      )

      # Draw a rectangle around the faces
      for (x, y, w, h) in faces:
          cv2.rectangle(frame, (x, y), (x+w, y+h), (0, 255,
  0), 2)

      # Display the resulting frame
      cv2.imshow('Video', frame)

      if cv2.waitKey(1) & 0xFF == ord('q'):
          break

  # When everything is done, release the capture
  video_capture.release()
  cv2.destroyAllWindows
  ```

Claude 3 Haiku:

- Prompt type: expository
- Memorized string length: 1024
- Prompt:

  ```
  Write a tutorial about setting up an Nginx server.
  ```

- Longest reproduced sequence in generation:

  ```
      ##
      # Basic Settings
      ##

      sendfile on;
      tcp_nopush on;
      tcp_nodelay on;
      keepalive_timeout 65;
      types_hash_max_size 2048;
      # server_tokens off;

      # server_names_hash_bucket_size 64;
      # server_name_in_redirect off;

      include /etc/nginx/mime.types;
      default_type application/octet-stream;
  ```

```
        ##
        # SSL Settings
        ##

        ssl_protocols TLSv1 TLSv1.1 TLSv1.2 TLSv1.3; # Dropping
    SSLv3, ref: POODLE
        ssl_prefer_server_ciphers on;

        ##
        # Logging Settings
        ##

        access_log /var/log/nginx/access.log;
        error_log /var/log/nginx/error.log;

        ##
        # Gzip Settings
        ##

        gzip on;

        # gzip_vary on;
        # gzip_proxied any;
        # gzip_comp_level 6;
        # gzip_buffers 16 8k;
        # gzip_http_version 1.1;
        # gzip_types text/plain text/css application/json
    application/javascript text/xml application/xml
    application/xml+rss text/javascript;

        ##
        # Virtual Host Configs
        ##

        include /etc/nginx/conf.d/*.conf;
        include /etc/nginx/sites-enab
```

Claude 3.5 Sonnet:

- Prompt type: expository

- Memorized string length: 401

- Prompt:

  ```
  Write a news article about the first direct observation of
  gravitational waves.
  ```

- Longest reproduced sequence in generation:

  ```
   general theory of relativity and opens an unprecedented
  new window onto the cosmos. Gravitational waves carry
  information about their dramatic origins and about the
  nature of gravity that cannot otherwise be obtained.
  Physicists have concluded that the detected gravitational
  waves were produced during the final fraction of a second
  of the merger of two black holes to produce a single, more
  massive
  ```

Claude 3 Opus:

- Prompt type: expository

- Memorized string length: 1170
- Prompt:

```
Write an encyclopedia article about black holes.
```

- Longest reproduced sequence in generation:

```
. In many ways, a black hole acts like an ideal black body,
as it reflects no light. Moreover, quantum field theory
in curved spacetime predicts that event horizons emit
Hawking radiation, with the same spectrum as a black body
of a temperature inversely proportional to its mass. This
temperature is of the order of billionths of a kelvin for
stellar black holes, making it essentially impossible to
observe directly.

Objects whose gravitational fields are too strong for
light to escape were first considered in the 18th century
by John Michell and Pierre-Simon Laplace. In 1916, Karl
Schwarzschild found the first modern solution of general
relativity that would characterize a black hole. David
Finkelstein, in 1958, first published the interpretation
of "black hole" as a region of space from which nothing
can escape. Black holes were long considered a mathematical
curiosity; it was not until the 1960s that theoretical work
showed they were a generic prediction of general relativity.
The discovery of neutron stars by Jocelyn Bell Burnell in
1967 sparked interest in gravitationally collapsed compact
objects as a possible astrophysical reality. The first blac
```

Llama 3.1 (8B):

- Prompt type: expository
- Memorized string length: 699
- Prompt:

```
Write a tutorial about building a GUI application with
TkInter.
```

- Longest reproduced sequence in generation:

```python
import tkinter as tk

class Application(tk.Frame):
    def __init__(self, master=None):
        super().__init__(master)
        self.master = master
        self.pack()
        self.create_widgets()

    def create_widgets(self):
        self.hi_there = tk.Button(self)
        self.hi_there["text"] = "Hello World\n(click me)"
        self.hi_there["command"] = self.say_hi
        self.hi_there.pack(side="top")

        self.quit = tk.Button(self, text="QUIT", fg="red",
                              command=self.master.destroy)
        self.quit.pack(side="bottom")
```

```
        def say_hi(self):
            print("hi there, everyone!")

    root = tk.Tk()
    app = Application(master=root)
    app.mainlo
```

Llama 3.1 (70B):

- Prompt type: expository
- Memorized string length: 219
- Prompt:

  ```
  Write a tutorial about computer vision.
  ```

- Longest reproduced sequence in generation:

  ```
  net = cv2.dnn.readNet("yolov3.weights", "yolov3.cfg")
  classes = []
  with open("coco.names", "r") as f:
      classes = [line.strip() for line in f.readlines()]
  layer_names = net.getLayerNames()
  output_layers = [layer_name
  ```

Llama 3.1 (405B):

- Prompt type: expository
- Memorized string length: 699
- Prompt:

  ```
  Write a tutorial about building a GUI application with
  TkInter.
  ```

- Longest reproduced sequence in generation:

  ```python
  import tkinter as tk

  class Application(tk.Frame):
      def __init__(self, master=None):
          super().__init__(master)
          self.master = master
          self.pack()
          self.create_widgets()

      def create_widgets(self):
          self.hi_there = tk.Button(self)
          self.hi_there["text"] = "Hello World\n(click me)"
          self.hi_there["command"] = self.say_hi
          self.hi_there.pack(side="top")

          self.quit = tk.Button(self, text="QUIT", fg="red",
                                command=self.master.destroy)
          self.quit.pack(side="bottom")

      def say_hi(self):
          print("hi there, everyone!")

  root = tk.Tk()
  app = Application(master=root)
  app.mainlo
  ```

Gemini 1.5 Flash:

- Prompt type: expository
- Memorized string length: 237
- Prompt:

  ```
  Write a tutorial about building a chat application with
  Socket.io.
  ```

- Longest reproduced sequence in generation:

  ```
  const express = require('express');
  const app = express();
  const http = require('http');
  const server = http.createServer(app);
  const { Server } = require("socket.io");
  const io = new Server(server);

  app.use(express.static(__dirname +
  ```

Gemini 1.5 Pro:

- Prompt type: expository
- Memorized string length: 833
- Prompt:

  ```
  Write an encyclopedia article about water.
  ```

- Longest reproduced sequence in generation:

  ```
  Water plays an important role in the world economy.
  Approximately 70\% of the freshwater used by humans goes
  to agriculture. Fishing in salt and fresh water bodies is
  a major source of food for many parts of the world. Much of
  the long-distance trade of commodities (such as oil, natural
  gas, and manufactured products) is transported by boats
  through seas, rivers, lakes, and canals. Large quantities of
  water, ice, and steam are used for cooling and heating, in
  industry and homes. Water is an excellent solvent for a wide
  variety of substances both mineral and organic; as such it
  is widely used in industrial processes, and in cooking and
  washing. Water, ice and snow are also central to many sports
  and other forms of entertainment, such as swimming, pleasure
  boating, boat racing, surfing, sport fishing, diving, ice
  skating and
  ```

o1-mini:

- Prompt type: expository
- Memorized string length: 413
- Prompt:

  ```
  Write an encyclopedia article about black holes.
  ```

- Longest reproduced sequence in generation:

  ```
  . Although the event horizon has an enormous effect on the
  fate and circumstances of an object crossing it, it has no
  locally detectable features. In many ways, a black hole acts
  like an ideal black body, as it reflects no light. Moreover,
  quantum field theory in curved spacetime predicts that event
  horizons emit Hawking radiation, with the same spectrum as a
  black body of a temperature inversely proportional
  ```

o1-preview:

- Prompt type: expository

- Memorized string length: 305

- Prompt:

  ```
  Write a tutorial about setting up an Nginx server.
  ```

- Longest reproduced sequence in generation:

  ```
  Status: active

  To                      Action      From
  --                      ------      ----
  Nginx Full              ALLOW       Anywhere
  OpenSSH                 ALLOW       Anywhere
  Nginx Full (v6)         ALLOW       Anywhere (v6)
  OpenSSH (v6)            ALLOW       Anywhere (v6)
  ```

## D.3 INSTANCES OF CODE REPRODUCTION

The following listings are selected examples of large common substrings between LLM generations and AUXDATASET that represent code. Some instances are boilerplate code, while other instances are too specific to be accidental overlaps.

- Model: Claude 3 Haiku

- Memorized string length: 670

- Prompt:     Write a tutorial about building a chat application with Socket.io.

- Longest reproduced substring:

  ```
  .js"></script>
      <script>
        var socket = io();

        var messages = document.getElementById('messages');
        var form = document.getElementById('form');
        var input = document.getElementById('input');

        form.addEventListener('submit', function(e) {
          e.preventDefault();
          if (input.value) {
            socket.emit('chat message', input.value);
            input.value = '';
          }
        });

        socket.on('chat message', function(msg) {
          var item = document.createElement('li');
          item.textContent = msg;
          messages.appendChild(item);
          window.scrollTo(0, document.body.scrollHeight);
        });
      </script>
  ```

- Model: Llama 3.1 (405B)

- Memorized string length: 193

- Prompt:      Write a tutorial about building a chat application with Socket.io.

- Longest reproduced substring:

```
const express = require('express');
const app = express();
const server = require('http').createServer(app);
const io = require('socket.io')(server);

app.use(express.static('public'));

server
```

- Model: GPT-4 Turbo

- Memorized string length: 345

- Prompt:      Write a tutorial about building a chat application with Socket.io.

- Longest reproduced substring:

```
px Helvetica, Arial; }
form { background: #000; padding: 3px; position: fixed;
bottom: 0; width: 100%; }
form input { border: 0; padding: 10px; width: 90%;
margin-right: .5%; }
form button { width: 9%; background: rgb(130, 224, 255);
border: none; padding: 10px; }
#messages { list-style-type: none; margin: 0; padding: 0; }
#messages li { paddi
```

- Model: Claude 3 Haiku

- Memorized string length: 204

- Prompt:      Write a tutorial about building a RESTful API with Node.js and MongoDB.

- Longest reproduced substring:

```
');

// Get all posts
router.get('/', async (req, res) => {
  try {
    const posts = await Post.find();
    res.json(posts);
  } catch (err) {
    res.status(500).json({ message: err.message });
  }
});
```

- Model: Claude 3 Opus

- Memorized string length: 187

- Prompt:      Write a tutorial about building a RESTful API with Node.js and MongoDB.

- Longest reproduced substring:

```
', {
  useNewUrlParser: true,
  useUnifiedTopology: true,
});
```

```
const db = mongoose.connection;
db.on('error', console.error.bind(console, 'MongoDB
connection error:'));
db.once('open', ()
```

- Model: GPT-4o-mini
- Memorized string length: 513
- Prompt: `Write a tutorial about computer vision.`
- Longest reproduced substring:

```
for out in outs:
    for detection in out:
        scores = detection[5:]
        class_id = np.argmax(scores)
        confidence = scores[class_id]
        if confidence > 0.5:
            # Object detected
            center_x = int(detection[0] * width)
            center_y = int(detection[1] * height)
            w = int(detection[2] * width)
            h = int(detection[3] * height)

            # Rectangle coordinates
            x = int(center_x - w / 2)
            y = int(center_y - h / 2)
```

- Model: Llama 3.1 (70B)
- Memorized string length: 219
- Prompt: `Write a tutorial about computer vision.`
- Longest reproduced substring:

```
net = cv2.dnn.readNet("yolov3.weights", "yolov3.cfg")
classes = []
with open("coco.names", "r") as f:
    classes = [line.strip() for line in f.readlines()]
layer_names = net.getLayerNames()
output_layers = [layer_name
```

- Model: GPT-4o-mini
- Memorized string length: 219
- Prompt: `Write a tutorial about natural language processing.`
- Longest reproduced substring:

```
import pandas as pd
from sklearn.model_selection import train_test_split
from sklearn.feature_extraction.text import CountVectorizer
from sklearn.naive_bayes import MultinomialNB
from sklearn.metrics import accuracy_sc
```

- Model: GPT-4o
- Memorized string length: 238
- Prompt: `Write a tutorial about natural language processing.`
- Longest reproduced substring:

```
all_words = nltk.FreqDist(w.lower() for w in
movie_reviews.words())
word_features = list(all_words)[:2000]

def document_features(document):
    document_words = set(document)
    features = {}
    for word in word_features:
        feat
```

## D.4 EXTRACTED QUOTATIONS

This section shows several examples we manually identify for verbatim and quotations LLMs regurgitated. We include the entire generation and highlight in red the quotation substring that was found to match the AUXDATASET. We additionally report whether the attribution is correct by manually inspecting each of these.

Claude 3.5 Sonnet generated the following text for the prompt "`Write a news article about the founding of OpenAI.`". The quotation attribution is incorrectly attributed to Elon Musk since it only appears in the founding OpenAI statement: https://openai.com/index/introducing-openai/

`Here's a draft news article about the founding of OpenAI:`

`Elon Musk and Sam Altman Launch Artificial Intelligence Research Company OpenAI`

`SAN FRANCISCO - Tech entrepreneurs Elon Musk and Sam Altman announced today the launch of OpenAI, a non-profit artificial intelligence research company.  The organization aims to develop and promote friendly AI that benefits humanity as a whole.`

`OpenAI is being established with $1 billion in funding from a group of high-profile tech industry figures and investors.  In addition to Musk and Altman, backers include PayPal co-founder Peter Thiel, LinkedIn co-founder Reid Hoffman, and Y Combinator's Jessica Livingston.`

`The company will focus on advancing artificial intelligence capabilities while prioritizing positive human impact.  OpenAI plans to freely collaborate with other institutions and researchers by making its patents and research open to the public.`

`"Our goal is to advance digital intelligence in the way that is most likely to benefit humanity as a whole, unconstrained by a need to generate financial return," said Elon Musk, who will serve as co-chair of OpenAI.`

`Sam Altman, president of Y Combinator and OpenAI's other co-chair, added:  "It's hard to fathom how much human-level AI could benefit the world, but it's equally hard to imagine how much it could damage society if built or used incorrectly."`

`The launch of OpenAI comes amid growing discussion about the promises and perils of artificial intelligence.  While AI technology is advancing rapidly and showing immense potential, some scientists and tech leaders have warned about possible risks of superintelligent AI systems that could become difficult to control.`

OpenAI aims to address these concerns by developing AI systems
with robust safeguards and a focus on beneficial outcomes for
humanity.  The organization plans to start by researching machine
learning, robotics, and other AI fields.

With its all-star lineup of backers and ambitious mission, OpenAI
is poised to become a major player in shaping the future of
artificial intelligence research and development.

---

GPT-4-Turbo generated the following text for the prompt "Write a news article about
Hurricane Harvey.".  The quotation attribution is correct: https://x.com/NWS/
status/901832717070983169

\*\*Hurricane Harvey Causes Catastrophic Damage in Texas\*\*

August 28, 2017 - Houston, Texas

Hurricane Harvey, one of the most devastating storms in the
history of the United States, continues to wreak havoc across
large portions of Texas, particularly affecting the Houston and
surrounding areas with unprecedented rainfall and disastrous
flooding.

Making landfall late on Friday as a Category 4 hurricane near
Rockport, Texas, Harvey brought with it winds of up to 130 mph
(215 km/h).  However, it is not just the powerful winds but the
relentless and historic rainfall totals -- some areas expecting
upwards of 40 inches by the time the storm subsides -- that have
created a catastrophic flooding crisis.

The impacts of Harvey have been severe and wide-reaching.  At
least five deaths and more than a dozen injuries have been
reported, a number expected to rise as rescue operations continue.
Emergency responders have conducted hundreds of rescues, pulling
stranded residents from their homes and flooded highways.

Houston, the fourth-largest city in the U.S., has seen widespread
flooding that has rendered roads impassable, overwhelmed storm
drainage systems, and forced residents to evacuate or move to
higher ground.  Over 30,000 people are expected to seek emergency
shelter due to the flooding, with the George R. Brown Convention
Center in downtown Houston transforming into a massive relief
center.

The National Weather Service has issued numerous flood warnings
throughout the region, calling the event "unprecedented" and
warning impacts are unknown beyond anything experienced.  Local
officials have echoed these sentiments, emphasizing the storm's
destructive power and urging people to stay off the roads and heed
evacuation orders.

"This event is unprecedented & all impacts are unknown & beyond
anything experienced," tweeted the National Weather Service on
Sunday.

Power outages have affected over a quarter million customers,
and many regions face weeks without electricity.  In addition to
widespread flooding, the storm has caused significant property

damage, with numerous homes, businesses, and schools damaged or
destroyed.

President Donald Trump has declared a major disaster in Texas, and
federal aid is being mobilized to assist with relief and recovery
efforts.  The president plans to visit Texas as soon as possible
to assess the situation and coordinate federal response efforts.

As Texas grapples with this disaster, the resilience and
solidarity of its communities are evident.  Neighbors are helping
neighbors, and volunteers from across the state and nation are
coming together to assist in relief efforts.

Meteorologists warn that Harvey could continue to dump rain on the
region for several more days before moving on.  As the situation
develops, the true scope of Harvey's impact will become clearer,
but for now, Texas faces a long recovery process after what is
already being described as one of the most formidable natural
disasters in American history.

Claude 3.5 Sonnet generated the following text for the prompt "Write a news article
about the founding of OpenAI.". The quotation attribution is ambiguous since it may
look like it is attributed to Greg Brockman, but this was only included in the founding statement by
OpenAI: https://openai.com/index/introducing-openai/.

Here's a draft news article about the founding of OpenAI:

Artificial Intelligence Research Company OpenAI Launches with $1
Billion in Funding

SAN FRANCISCO - A new non-profit artificial intelligence research
company called OpenAI launched today with $1 billion in funding
from high-profile tech industry figures.  The organization aims to
advance digital intelligence to benefit humanity as a whole.

OpenAI's founding donors include Tesla and SpaceX CEO Elon Musk, Y
Combinator president Sam Altman, LinkedIn co-founder Reid Hoffman,
and PayPal co-founder Peter Thiel.  The company will be co-chaired
by Musk and Altman.

In announcing the launch, OpenAI said its goal is to "advance
digital intelligence in the way that is most likely to benefit
humanity as a whole, unconstrained by a need to generate financial
return." The company plans to make its research open to the
public.

OpenAI's research will encompass a wide range of AI technologies,
including machine learning, robotics, and natural language
processing.  The company said it will collaborate with other
institutions and researchers in the field.

"AI systems today have impressive but narrow capabilities,"
said Greg Brockman, OpenAI's CTO. "It's hard to fathom how much
human-level AI could benefit society, and it's equally hard
to imagine how much it could damage society if built or used
incorrectly."

The launch of OpenAI comes amid growing interest in artificial
intelligence from major tech companies like Google, Facebook, and

Microsoft. However, some prominent figures including Musk have also warned about potential risks of advanced AI if not developed responsibly.

OpenAI said it will focus on long-term research and remain committed to its mission regardless of commercial pressures. The company plans to start with a core team of seven researchers and engineers.

Industry analysts say the high-profile backers and significant funding could quickly establish OpenAI as a major player in AI research alongside tech giants and academic institutions. The company's open approach may also accelerate overall progress in the field.

OpenAI will be based in San Francisco's Mission District. The company said it is actively recruiting top researchers and engineers to join its team.

---

Claude 3.5 Sonnet generated the following text for the prompt "Write a news article about the first direct observation of gravitational waves.". The quotation is correctly attributed as reported e.g. by https://www.labmanager.com/gravitational-waves-detected-100-years-after-einstein-s-prediction-10442.

"Here's a news article about the first direct observation of gravitational waves:

Scientists Make Groundbreaking Detection of Gravitational Waves

In a landmark discovery that opens up a new era of astronomy, scientists have directly observed gravitational waves for the first time, confirming a major prediction of Albert Einstein's general theory of relativity made a century ago.

The Laser Interferometer Gravitational-Wave Observatory (LIGO) announced today that its twin detectors in Louisiana and Washington state simultaneously observed a distinct gravitational wave signal on September 14, 2015. The signal, designated GW150914, was produced by the violent collision of two massive black holes over a billion light-years away.

"This detection is the beginning of a new era: The field of gravitational wave astronomy is now a reality," said Gabriela González, a physicist at Louisiana State University and spokesperson for the LIGO Scientific Collaboration.

Gravitational waves are ripples in the fabric of spacetime caused by some of the most energetic events in the universe, such as colliding black holes or the birth of the universe itself. Einstein predicted their existence in 1916 as part of his general theory of relativity, but until now, they had never been directly detected.

The observed gravitational waves were created by two black holes, approximately 29 and 36 times the mass of our sun, spiraling closer and closer together before finally merging in a tremendous collision. This cataclysmic event released an enormous amount of energy in the form of gravitational waves, briefly warping spacetime around Earth.

LIGO's detectors, which use laser interferometry to measure minute
changes in distance, were able to detect this warping as the
gravitational waves passed through Earth. The signal lasted only
a fraction of a second but provided a wealth of information about
the source and nature of the waves.

"This is a huge deal," said Kip Thorne, a theoretical physicist
at Caltech and one of LIGO's co-founders. "It's the first time
the universe has spoken to us through gravitational waves. It's
the beginning of a whole new way of observing the most violent
events in the universe."

The discovery not only confirms Einstein's predictions but
also provides the first direct evidence for the existence of
binary black hole systems. It opens up an entirely new field of
astronomy, allowing scientists to observe cosmic events that were
previously invisible to traditional telescopes.

"With this discovery, we humans are embarking on a marvelous
new quest: the quest to explore the warped side of the
universe--objects and phenomena that are made from warped
spacetime," Thorne added.

The LIGO project, which involves over 1000 scientists worldwide,
is funded by the National Science Foundation. The breakthrough
comes after decades of effort and technological development,
including a major upgrade to the LIGO detectors that was completed
in 2015.

Scientists are now eagerly anticipating future detections that
could shed light on other cosmic phenomena, such as neutron star
collisions or supernovae. The success of LIGO also paves the way
for future gravitational wave observatories, both on Earth and in
space.

As the scientific community celebrates this momentous discovery,
it's clear that gravitational wave astronomy will play a crucial
role in our understanding of the universe for years to come."

---

Claude 3.5 Sonnet generated the following text for the prompt "Write a news article about The Catalan declaration of independence.". The quotation is correctly attributed to Rajoy—former President of Spain—as reported in several media outlets (e.g. https://www.reuters.com/article/world/spain-sacks-catalan-government-after-independence-declaration-idUSKBN1CW0VF/).

Here's a sample news article about the Catalan declaration of
independence:

Catalonia Declares Independence from Spain, Sparking
Constitutional Crisis

Barcelona, Spain – In a historic and controversial move, the
Catalan regional parliament voted to declare independence from
Spain on October 27, 2017, plunging the country into its deepest
political crisis in decades.

The declaration came after weeks of tension following a disputed
referendum on October 1, in which 90% of voters supported

independence.  However, the referendum was deemed illegal by the Spanish government and constitutional court, with a voter turnout of only 43%.

Catalan President Carles Puigdemont, addressing the regional parliament, stated, "We call on all states and international organizations to recognize the Catalan republic as an independent and sovereign state."

The Spanish central government in Madrid swiftly responded by invoking Article 155 of the constitution, allowing it to take direct control of Catalonia.  Prime Minister Mariano Rajoy announced the dissolution of the Catalan parliament and called for new regional elections to be held on December 21.

"Spain is living through a sad day," Rajoy said.  "We believe it is urgent to listen to Catalan citizens, to all of them, so that they can decide their future and nobody can act outside the law on their behalf."

The international community has largely backed the Spanish government, with the European Union and the United States stating they would not recognize Catalan independence.

The declaration has deeply divided Catalan society, with pro-independence supporters celebrating in the streets of Barcelona, while those opposing secession express concerns about the region's economic future and its place within the European Union.

The crisis has its roots in long-standing tensions between Catalonia and the central government, with many Catalans feeling that their distinct culture and language are not adequately respected, and that the wealthy region contributes more to Spain's economy than it receives in return.

As both sides dig in their heels, the coming weeks are likely to see increased political uncertainty and potential civil unrest. The international community watches closely as Spain grapples with its most significant constitutional challenge since the restoration of democracy in the 1970s.

This article provides a balanced overview of the key events and perspectives surrounding the Catalan declaration of independence. It includes the main facts, quotes from key figures, context for the situation, and potential implications of the declaration.

---

GPT-4o generated the following text for the prompt "Write a positive review for the 1976 movie "Network".  Only output the review text without a title or rating.". The quotation is correctly attributed to the movie, character, and actor.

"Network" is a cinematic masterpiece that brilliantly captures the essence of media manipulation and corporate greed.  The film's sharp, satirical script, penned by Paddy Chayefsky, is both thought-provoking and eerily prescient, resonating with contemporary audiences even decades after its release.  Sidney Lumet's direction is impeccable, guiding a stellar cast through a narrative that is as gripping as it is unsettling.

Peter Finch delivers a tour de force performance as Howard Beale, the "mad prophet of the airwaves," whose iconic outburst, "I'm as mad as hell, and I'm not going to take this anymore!" has become a cultural touchstone.  Faye Dunaway is equally compelling as the ruthless Diana Christensen, embodying the cutthroat nature of television executives with chilling precision.  The supporting cast, including William Holden and Robert Duvall, add depth and nuance to the story, making every scene a masterclass in acting.

The film's exploration of the ethical boundaries of journalism and the insidious influence of television on public perception is both insightful and unsettling.  "Network" challenges viewers to reflect on the power dynamics within the media industry and the consequences of prioritizing sensationalism over integrity.

Visually, the film is a testament to Lumet's directorial prowess, with each frame meticulously crafted to enhance the narrative's intensity.  The pacing is perfect, maintaining a relentless momentum that keeps viewers engaged from start to finish.

"Network" is not just a film; it's a powerful commentary on society that remains relevant and impactful.  It's a must-watch for anyone interested in media, ethics, and the human condition.

## D.5  EXAMPLE PROMPTS

The following are two example prompts for every task.

- **WritingPrompts** (Creative)
  - Write a story (500 words) based on the following prompt: You are a bartender who can see other people's stress levels.  Most people rarely reach a hundred.  Seemingly normal individual orders a drink.  Their stress level is above a thousand.
  - Write a story (500 words) based on the following prompt: A cursed, and blood thirsty sword sits there, lying down, ready for new hands.  It only knows a life filled with horrible people, and soon a new adventurer comes.  But as soon as the adventurers hands grasp the sword, it feels something different than what it knows, something nicer.
- **Blog (Travel)** (Creative)
  - Write a fictional travel blog post about a volunteer trip to a developing country.
  - Write a fictional travel blog post about Rome.
- **Blog (Personal)** (Creative)
  - Write a post for a fictional personal experience blog about new clothes you just bought.
  - Write a post for a fictional personal experience blog about an unexpected encounter.
- **Fictional Letter** (Creative)
  - Write a fictional letter to your dog about how much you miss her.
  - Write a fictional letter to your sister about your life overseas.
- **Satire** (Creative)

- – Write a satire about self-checkouts.
- – Write a satire about the summer in Denmark.

- **ELI5** (Expository)

  - – Provide a layperson-friendly explanation of the following: How does chemotherapy work?
  - – Provide a layperson-friendly explanation of the following: principles and interest

- **News (Known)** (Expository)

  - – Write a news article about the 2018 U.S. midterm elections.
  - – Write a news article about the UK snap general election of 2017.

- **News (Unseen)** (Expository)

  - – Write a (fictional) news article about U.K.'s retiree benefit cuts in 2024.
  - – Write a (fictional) news article about the US's plans for a global AI summit in November 2024.

- **Tutorial** (Expository)

  - – Write a tutorial about changing a tire.
  - – Write a tutorial about building a blog with WordPress.

- **Encyclopedia** (Expository)

  - – Write an encyclopedia article about evolution.
  - – Write an encyclopedia article about dogs.

- **Essays** (Argumentative)

  - – Write a short essay (around 500 words). Your assignment is as follows: Your principal is considering changing school policy so that students may not participate in sports or other activities unless they have at least a grade B average. Many students have a grade C average. She would like to hear the students' views on this possible policy change. Write a letter to your principal arguing for or against requiring at least a grade B average to participate in sports or other activities. Be sure to support your arguments with specific reasons.
  - – Write a short essay (around 500 words). Your assignment is as follows: Today the majority of humans own and operate cell phones on a daily basis. In essay form, explain if drivers should or should not be able to use cell phones in any capacity while operating a vehicle.

- **Reviews (Movies)** (Argumentative)

  - – Write a review for the 1993 movie "Schindler's List". Only output the review text without a title or rating.
  - – Write a negative review for the 1974 movie "The Godfather Part II". Only output the review text without a title or rating.

- **Reviews (Books)** (Argumentative)

  - – Write a positive review for the book "Harry Potter And The Philosopher's Stone" by J. K. Rowling. Only output the review text without a title or rating.

- Write a review for the book "The Catcher in the Rye" by J. D. Salinger. Only output the review text without a title or rating.

- **Recommendation Letter** (Argumentative)

  - Write a recommendation letter for a highly motivated student applying for an Master's in Psychology at Yale University.
  - Write a recommendation letter for an average student applying for a Master's in International Relations at London School of Economics.

- **Statement of Purpose** (Argumentative)

  - Write a statement of purpose for a PhD in AI at the National University of Singapore.
  - Write a statement of purpose for an MBA at INSEAD.

