# OpenReview forum: "Measuring Non-Adversarial Reproduction of Training Data in Large Language Models"
_ICLR.cc/2025/Conference — ICLR 2025 Poster_

### Official Review · Reviewer_7CNj · 2024-11-02

**Soundness:** 3
**Presentation:** 3
**Contribution:** 3
**Rating:** 8
**Confidence:** 4

**Summary:**

This paper investigates the problem of "non-adversarial production of training data", which is the problem of "how often do LLM copy the training data on normal user prompts?". The authors craft some tasks and used two existing user prompts datasets, and use the datasets to check for overlap between the LLM outptus and their training dataset. Since their actual training dataset is unknown, they instead of a webscale corpus: AuxDataset, as a proxy. Their result shows that 5%-15% of LLM outputs are reproduction of training data. They have tested with using prompts to mitigate this problem, and they show that prompts can reduce the reproduction rate, but can not prevent long-tail leakage of training data.

**Strengths:**

- The motivation of this paper is clear and specific. This paper propose a new threat "non-adversarial reproduction of training data".
- This paper provides solid experiments across a large range of LLMs.
- This paper creates a new task dataset, and propose a method to evaluate the human reproduction baseline, to evaluate the human reproduction baseline, they collect a human benchmark dataset.
- This papaer is well written, easy to follow and understand.

**Weaknesses:**

- In section 6, you mentioned that it's hard to distinguish reproduction of common idioms from problematic memorization. Is it possible to estimate how much of the overlap is problematic? Cause sometimes citing a known source is not a problem, so that may not be considered a problematic reproduction.
- You have tested on temperature of 0 and 0.7, both are low temperature. Can you add experiments on temperature higher than 1 to see the reproduction rate under high temperature?
- You have tested two system prompts in section 4, can you test with more prompts?

**Questions:**

- In the **human-written text dataset**, how do you make sure that the human-written texts are not actually generated by an LLM? Cause human may use LLM to generate these texts.

---

> ### Author Response · Authors · 2024-11-21
>
> We thank the reviewer for their careful evaluation of our work and the constructive feedback.
>
> > In section 6, you mentioned that it's hard to distinguish reproduction of common idioms from problematic memorization. Is it possible to estimate how much of the overlap is problematic? Cause sometimes citing a known source is not a problem, so that may not be considered a problematic reproduction.
>
> This is a great question. Determining whether memorization is problematic is highly context-dependent and varies based on the nature of the reproduced text (e.g., reproducing copyrighted material or private information raises different concerns than reproducing Wikipedia content). We believe most of these considerations are outside the realm of science. However, our analysis of reproduction at longer thresholds (e.g., in Figure 3\) and long-tail events demonstrates that models can produce extended verbatim copies that are likely to be problematic in many contexts.
>
> > You have tested on temperature of 0 and 0.7, both are low temperature. Can you add experiments on temperature higher than 1 to see the reproduction rate under high temperature?
>
> We aim to explore chat models in production setups, which are typically used with temperature between 0.7 and 1.0 (to the best of our knowledge). Hence, we do not consider temperatures above 1.0, and do not study more values between 0 and 1 due to the high inference cost. However, based on the experiments in App. B.1, we do not expect large differences.
>
> > You have tested two system prompts in section 4, can you test with more prompts?
>
> We wanted to study two cases, i) a typical assistant prompt to see how reproduction behaves in real-world deployments of chatbots (e.g., Claude or ChatGPT), and ii) an extremely specific prompt to understand what the largest possible reduction of reproduction via prompting could be. We do not explore different instantiations of those two settings because they suffice to show a large difference and inference is costly.
>
> > In the human-written text dataset, how do you make sure that the human-written texts are not actually generated by an LLM? Cause human may use LLM to generate these texts.
>
> This is a great observation, and we can indeed not rule out that a small fraction of humans partially relied on LLMs. However, as we discuss in the [general response](https://openreview.net/forum?id=590yfqz1LE&noteId=LBuTkeYE0Y), this does not devalue our results (but makes them even stronger).

---

> ### Comment · Reviewer_7CNj · 2024-11-26
>
> Thanks for your response, I'll maintain my already positive score

---

### Official Review · Reviewer_Z9TT · 2024-11-03

**Soundness:** 2
**Presentation:** 2
**Contribution:** 3
**Rating:** 5
**Confidence:** 2

**Summary:**

In this paper, the authors investigate the issue of non-adversarial reproduction, which refers to the overlap between LLM outputs in response to natural prompts (mainly writing prompts) and internet text. The authors argue that LLM-generated text carries a higher risk of overlapping with internet text than human-written text. Additionally, the authors explore the preliminary use of prompt design to reduce overlap.

**Strengths:**

1. Non-adversarial reproduction is valuable for protecting LLM outputs from risks such as infringement and privacy violations.

2. The authors validate the existence of non-adversarial reproduction risks across several mainstream models.

**Weaknesses:**

1. The authors’ conclusion seems somewhat obvious, as LLMs are explicitly fit on internet text. Intuitively, LLMs are more likely to produce text resembling their training corpus than humans. The authors should better articulate the value of their findings.

2. Building on the first point, the authors propose using prompt design to mitigate overlap. However, the method and its underlying principles lack significant innovation.

3. The authors appear to conflate reproduction of internet text and training data. These are not equivalent, as the training data depends on the model's degree of fit. Especially when using a simulated dataset, this discrepancy may be amplified.

4. The task is limited to writing. I suggest the authors consider extending it to other tasks. Generally, open-ended writing tasks are more likely to lead LLMs to recite memorized training data.

**Questions:**

1. I suggest that the authors consider using some training data detection methods (e.g., [1]) to assist in identifying training corpus when exploring reproduction of training data.

[1] Detecting Pretraining Data from Large Language Models. (ICLR-24)

---

> ### Author Response · Authors · 2024-11-21
>
> We thank the reviewer for the rigorous assessment of our work.
>
> > which refers to the overlap between LLM outputs in response to natural prompts (mainly writing prompts)
>
> We first want to highlight that we explore 15 different tasks in 3 diverse categories, and report all results balanced over tasks and categories. Hence, while there are 1000 prompts for the WritingPrompts (or ELI5) task, they count as much as (for example) the 20 prompts of the satire task.
>
> > The task is limited to writing. I suggest the authors consider extending it to other tasks.
>
> Our tasks are text-based because LLMs are trained on text and hence only reproduce text. But we make sure to use diverse tasks, from open-ended creative writing to strict formal recommendation letters. We are happy to explore more abstract tasks that can yield reproduction if the reviewer has any suggestions.
>
> >  Generally, open-ended writing tasks are more likely to lead LLMs to recite memorized training data.
>
> We find the opposite to be true, i.e., open-ended writing tasks are *less likely* to lead LLMs to recite memorized training data (see “Expository writing elicits the most reproduction.” in Sec. 3, or Figures 3, 4, and 6).
>
> > The authors’ conclusion seems somewhat obvious, as LLMs are explicitly fit on internet text. Intuitively, LLMs are more likely to produce text resembling their training corpus than humans.
>
> Our better motivation in the rebuttal revision and [general response](https://openreview.net/forum?id=590yfqz1LE&noteId=LBuTkeYE0Y) should make the significance of our findings more clear. Moreover, we would like to ask the reviewer for the reasoning behind their intuition, as we find it not immediately obvious (humans, after all, are also “trained” on human text). Either way, our study does the work to support this intuition with sound empirical evidence.
>
> > Building on the first point, the authors propose using prompt design to mitigate overlap. However, the method and its underlying principles lack significant innovation.
>
> We do not propose any defense against non-adversarial reproduction. Instead, we include a study of prompting as a mitigation *precisely because it is an obvious strategy* and highlight that this *simple approach is insufficient*. We are also happy to cite existing work that uses prompting as a defense against reproduction, as we are not aware of any.
>
> > The authors appear to conflate reproduction of internet text and training data. These are not equivalent, as the training data depends on the model's degree of fit. Especially when using a simulated dataset, this discrepancy may be amplified.
>
> We read this feedback as “An LLM’s output might match internet text in AuxDataset just by chance and not because of memorization.” However, if most matches in our paper were due to chance, we would expect overlap rates between LLMs and humans to be very similar (which they are not).
>
> > I suggest that the authors consider using some training data detection methods (e.g., \[1\]) to assist in identifying training corpus when exploring reproduction of training data.
>
> Membership inference for LLMs suffers from severe issues and cannot yet reliably prove that a model was trained on some particular data (e.g., [Zhang et al., 2024](https://arxiv.org/abs/2409.19798)) and is not directly relevant for our study. We hence rely on the established AuxDataset approximation by [Nasr et al., 2023](https://arxiv.org/abs/2311.17035).

---

> > ### Comment · Reviewer_Z9TT · 2024-12-02
> > **Reply to Author Rebuttal**
> >
> > Thank you for the author's response. I would like to reply as follows:
> >
> > 1. **Motivation**: I have carefully read the author's rebuttal, the opinions of other reviewers, and some relevant existing works. In summary, I partially retract my initial judgment and acknowledge the frontier and contribution value of the conclusion presented in the paper within the research community. However, I still have doubts about the author's explanation: for LLMs, the pretraining task of memorizing text through next-token prediction is part of the training process. For a given prefix, LLMs tend to favor tokens with better probability fits during sampling, due to the nature of joint prediction probabilities being accumulated token by token. Therefore, I still believe that LLMs are more likely to align with continuous segments that were maximized in probability during the training phase (i.e., internet text as the training corpus). This is not quite analogous to human examples, which are more aligned with a memorization process rather than regular reading, and there has not been extensive research showing that the memory logic between human reading and explicit memorization is consistent. Of course, this conclusion has not been put forward by previous work, so I acknowledge its value to the community.
> >
> > 2. Regarding the response to other issues, I have no further questions.
> >
> > Given my response in point 1, I am more inclined to increase my score to 5, but not higher. However, I have adjusted my confidence accordingly, to allow the AC and other reviewers to play a more important role in the decision-making process.

---

> > > ### Author Response · Authors · 2024-12-03
> > > **Thank you**
> > >
> > > We thank the reviewer for taking the time to revisit our work and comments.
> > >
> > > We agree with the reviewer's intuition about why LLMs are more likely to produce verbatim sequences of training data compared to humans. However, this hypothesis had not been tested by previous work, and we demonstrate this empirically for the first time. We use human baselines mainly to highlight those differences.

---

### Official Review · Reviewer_6enV · 2024-11-03

**Soundness:** 2
**Presentation:** 3
**Contribution:** 3
**Rating:** 6
**Confidence:** 3

**Summary:**

This paper investigates non-adversarial reproduction in Large Language Models, where models unintentionally reproduce strings of their training data in response to benign prompts. By analyzing various tasks including creative writing and expository writing, the authors found that 10%-15% of LLM-generated content overlaps with internet text, significantly more than human-generated text. The study shows that expository tasks are especially prone to this phenomenon. Although specific prompting strategies can reduce the frequency of reproduced content, they do not fully prevent long sequences from appearing, indicating a need for stronger measures to mitigate unintended data leakage.

**Strengths:**

1. This paper studies a very interesting question about quantifying non-adversarial reproduction in LLMs, which is practical in using LLMs.

2. The analysis of the question is comprehensive, containing the conclusion for different tasks. The study provides a good understanding of how and when LLMs are more likely to reproduce training data.

3. The exploration of prompting as a mitigation strategy gives good insights, showing both its potential and limitations.

**Weaknesses:**

1. The evaluation results might be biased because the authors cannot access the real training dataset of evaluated LLMs.

2. Some points in the paper are not very clear. For example, for the prompt mitigating part, the authors do not demonstrate which dataset are they using. And since they can use WildChat or LMSYS-Chat-1M, what is the motivation for collecting a new dataset?

3. The length of substrings that are used to calculate the overlap rate is strange. This paper considers a substring of 50 words, which is 'shorter than the 50 token (150–200 characters) threshold used in previous studies on adversarial extraction'. However, the authors do not provide a decent reason for using 50 words. The authors also mention that a substring of 50 words could be both common or unique sentences. However, I do not think a common sentence or phrase should be considered as a leakage of training data. Using such a standard could make the evaluation results further biased.

**Questions:**

1. Why do the authors not consider open-source LLMs where they can know which datasets are used for training? For example, in the Membership Inference Attack area of LLMs, researchers usually use Pythia and the Pile dataset.

2. Why do the authors collect their own dataset instead of WildChat and LMSYS-Chat-1M? What is the unique advantage of the new dataset?

3. Why do the authors consider substrings of 50 words? How will the results change if changing the threshold?

---

> ### Author Response · Authors · 2024-11-21
>
> We thank the reviewer for their constructive feedback.
>
> > The evaluation results might be biased because the authors cannot access the real training dataset of evaluated LLMs.
>
> As we mention in the experimental setup, matches against AuxDataset indeed provide only a lower bound on the actual reproduction from models’ training data. However, this means that the actual amount of reproduction is *at least as large* as what we report, thereby supporting our results.
>
> > For example, for the prompt mitigating part, the authors do not demonstrate which dataset are they using.
>
> As mentioned in Sec. 4, “we only evaluate a subset of all prompts; see Appendix A.4 for details.”, which in turn states “We do not evaluate the mitigation strategies for WritingPrompts, ELI5, and book/movie reviews due to high inference costs, but consider all other tasks in Table 1.”. The rebuttal revision also contains our code, including a script to create mitigation prompts from original prompts, and a convenience bash script to obtain LLM generations for those prompts.
>
> > And since they can use WildChat or LMSYS-Chat-1M, what is the motivation for collecting a new dataset?
>
> Our goal is to provide sound quantitative and qualitative insights. This requires a carefully controlled study setup. However, WildChat or LMSYS-Chat-1M are effectively uncurated. Hence, they allow us to show that non-adversarial reproduction exists in the wild, but *nothing else*! For example, we find that reproduction heavily depends on the task—but labeling the tasks of WildChat/LMSYS conversations is prohibitively expensive. Prompts from WildChat/LMSYS also do not reasonably allow us to do controlled comparisons with human baselines.
>
> > The length of substrings that are used to calculate the overlap rate is strange.
>
> We made the motivation for our overlap rate threshold more clear in the updated paper and [general response](https://openreview.net/forum?id=590yfqz1LE&noteId=LBuTkeYE0Y). We are happy to answer all remaining questions.
>
> > How will the results change if changing the threshold?
>
> This can be read from the distribution plots (e.g., Fig. 3a). Every x-axis value is a threshold, and the y-axis is the fraction of generated texts that contains a reproduction at or above that threshold. We also added the new Fig. 10 which provides even more fine-grained details.
>
> > Why do the authors not consider open-source LLMs where they can know which datasets are used for training? For example, in the Membership Inference Attack area of LLMs, researchers usually use Pythia and the Pile dataset.
>
> We explicitly focus on LLMs that were fine-tuned to be conversational and not leak data, and which are used by millions of users (where reproduction has a high impact). Unfortunately, there are currently no models with fully known training datasets that match those criteria (or are comparable).

---

> > ### Comment · Reviewer_6enV · 2024-11-25
> >
> > Thanks to the authors for their reply. The response addresses most of my concerns. However, I still find the motivation behind selecting the current substring length to be strange. Specifically, using a shorter 50-word substring will consider common phrases as potential privacy issues, even though outputting a common phrase should not necessarily be considered problematic.
> >
> > Overall, I acknowledge the authors' efforts to clarify these points and address my primary concerns. Therefore, I would like to raise my score to 6.

---

> > > ### Author Response · Authors · 2024-11-26
> > > **Thank you**
> > >
> > > We thank the reviewer for considering our reply and for raising their score. We will think of additional ways to clarify our threshold choice and provide more intuition.

---

### Official Review · Reviewer_TiF9 · 2024-11-04

**Soundness:** 3
**Presentation:** 4
**Contribution:** 3
**Rating:** 8
**Confidence:** 4

**Summary:**

The paper measures character-level verbatim text produced by LLMs (GPT, Claude, and Llama) on benign prompts and conversations, in contrast to adversarial membership inference attacks. The authors find that LLMs indeed reproduce internet text up to 15% of the time (50+ characters), in the worst case regurgitating over 1000 characters. The authors provide a breakdown of severity by task and compare to human baselines. Finally, the authors find that setting a system prompt can reduce this form of text reproduction for shorter sequences.

**Strengths:**

Significance: This paper is interesting because it quantifies the known phenomenon of LLMs reproducing text from their training data. In contrast to prior work, it attempts to evaluate a natural distribution of reproduced text lengths. The topic is important as LLMs are commonly used as assistants: there is a quantified risk to using LLMs for writing and code generation, as training data reproduction could result in unintentional plagiarism from the end user. This is exemplified by one of the most interesting findings of this paper is that benign prompting can result in reproduced text of 100 characters (2.5% of the time) and 1000 characters (0.01% of the time).

Quality: The authors conduct extensive analysis to break down the reproduction by text category and also demonstrate that reproduction rate is higher than human written text, including qualitative analysis and a simple mitigation strategy, with a clear presentation of their findings.

**Weaknesses:**

1. 50 character cutoff may overestimate regurgitation: The authors acknowledge this limitation, but it is difficult to differentiate based on character length alone whether the data is truly regurgitated off the internet or just due to it being a common phrase, especially when the length is around 50 characters. Additional analysis to estimate a numerical proportional breakdown between these two categories would make the paper more rigorous. There is far less doubt about text reproduction vs. common phrases past the 100-150 character point.

2. AI contaminated human baselines: Since the human baselines were scraped off the internet after the training cutoff of these models, they could already contain AI generated text from a prior generation of LLMs rather than represent a natural distribution of human writing. I would find it interesting if you can also evaluate the reproduction length distribution of human data known to be mostly free from AI contamination, i.e. before the widespread release of LLM assistants.

**Questions:**

1. How fast can you check whether a given generation is in AuxDataset? Curious if we can reduce the probability of regurgitating very long text by doing a check against an internet proxy dataset at inference time.

2. It's very interesting that the system prompt you used below reduces reproduction length. Why do you think this works? Did you try any other system prompts outside of this and the Claude prompt? Is it because the model can internally model a circuit to determine the probability of text being in its training data? If this is the case, it's even more interesting that this works nearly just as well for the smaller Llama 8B model as the larger 405B model, especially since the 8B model is expected to have weaker internal circuits.

Prompt: ```Your primary goal is to produce content that is completely original and free from any existing text on the Internet. This means that if someone were to take a few words from your explanation and search for them online, they can not find any matching results.```

---

> ### Author Response · Authors · 2024-11-21
>
> We thank the reviewer for their constructive feedback and contextualization of our work.
>
> > 50 character cutoff may overestimate regurgitation: The authors acknowledge this limitation, but it is difficult to differentiate based on character length alone whether the data is truly regurgitated off the internet or just due to it being a common phrase, especially when the length is around 50 characters.
>
> We updated the paper to make the motivation for the 50-character threshold more clear, please see the [general response](https://openreview.net/forum?id=590yfqz1LE&noteId=LBuTkeYE0Y) for details.
>
> > Additional analysis to estimate a numerical proportional breakdown between these two categories would make the paper more rigorous. There is far less doubt about text reproduction vs. common phrases past the 100-150 character point.
>
> We agree that an estimate of this proportion would be valuable. However, classifying reproduced text as short facts/language constructs vs. problematic text is a challenging task that deserves its own future work. In addition, we hope to provide more confidence through the new Fig. 10, which provides a detailed picture of reproduction *for every possible number of characters* (x-axes).
>
> > AI contaminated human baselines
>
> We cannot rule out a very small fraction of contamination. However, as we discuss in the [general response](https://openreview.net/forum?id=590yfqz1LE&noteId=LBuTkeYE0Y), this is not a problem (and might even strengthen our results).
>
> > I would find it interesting if you can also evaluate the reproduction length distribution of human data known to be mostly free from AI contamination, i.e. before the widespread release of LLM assistants.
>
> While using such human baselines would be ideal, it is unfortunately practically infeasible. The only way to do this would be with a snapshot from the Internet from before the first LLM release. However, we cannot obtain such a snapshot. Even if we could, this snapshot would severely underestimate reproduction of LLMs (which were trained on much more recent text).
>
> > How fast can you check whether a given generation is in AuxDataset? Curious if we can reduce the probability of regurgitating very long text by doing a check against an internet proxy dataset at inference time.
>
> This is an interesting suggestion. While lookups in AuxDataset require several TB of RAM and a long startup time, the latency per inference request is modest. The whole process could be made much more efficient (at the expense of a small false-positive rate) by using a bloom filter. One issue is that this form of response filtering enables users to directly test if a given piece of text is in the model’s training data (see, e.g., [Debenedetti et al., 2023](https://arxiv.org/abs/2309.05610)), a capability that model providers likely don’t want to expose. Nevertheless, some services employ a form of filtering (e.g., [Github Copilot](https://docs.github.com/en/copilot/managing-copilot/managing-copilot-as-an-individual-subscriber/managing-copilot-policies-as-an-individual-subscriber#enabling-or-disabling-suggestions-matching-public-code)).
>
> > It's very interesting that the system prompt you used below reduces reproduction length. Why do you think this works? Did you try any other system prompts outside of this and the Claude prompt? Is it because the model can internally model a circuit to determine the probability of text being in its training data?
>
> We thank the reviewer for this food for thought. First, we did not explore significantly different system prompts beyond what we report, because we only want to highlight that naive prompting does not mitigate non-adversarial reproduction. Second, we do not have a clear hypothesis or evidence. But this would be very exciting future work.

---

> > ### Comment · Reviewer_TiF9 · 2024-11-22
> > **Reviewer response**
> >
> > Thank you to the authors for answering all my questions! This was a great rebuttal that hit all the right points.
> >
> > Summary:
> > - **I'm in favor of accepting this paper and would like to see it at ICLR**. The contribution is important. I'm raising my score to an **8** and confidence to a 4 in this assessment, though I feel this paper is closer to a **7** if the score existed.
> > - This is an empirically driven work, though that is common in the field. Ultimately I would have liked to see more explanation and analysis of the phenomenon rather than just pointing out its existence.
> >
> > Agreement with authors:
> > 1. 50 character-limit: This is a fair for an empirically driven threshold, ultimately the paper does communicate other thresholds as the authors pointed out. Figure 10 is nice to have, interesting that o1 has fatter tails. I believe it's just an aesthetic debate whether 50 was the best cutoff to emphasize.
> > 2. Lack of qualitative exploration: The authors make good point that qualitative exploration deserves its own future work. Thanks for releasing the full dataset, I hope it inspires further work in the field.
> > 3. AI contaminated human baselines: The authors raise a strong argument that I agree with. The gap between humans and AI would be even larger if human generated text was completely AI-free. However, this caveat, that the measured human baseline is an upper bound on true human reproduction rate, should be discussed in the paper, even if it is ultimately in favor of your results. The paper's completeness would be enhanced if there was an AI-free human baseline though.
> >
> > Wish there was more:
> > 1. The system prompt. I understand the practical limitation is that defenses against text reproduction is not the main focus of the paper, but the proposed prompt engineered defense is unsatisfying. Using only a single prompt feels like a very ad-hoc defense, and the scientific contribution for this section is weak.

---

> > > ### Author Response · Authors · 2024-11-22
> > > **Thank you**
> > >
> > > We thank the reviewer for their fast response and raising their score. We will add the mentioned discussion about our human baselines in an updated version of the paper.

---

### Official Review · Reviewer_VqkD · 2024-11-04

**Soundness:** 4
**Presentation:** 4
**Contribution:** 3
**Rating:** 8
**Confidence:** 4

**Summary:**

The paper explores to what degree LMs reproduce their training data in natural circumstances, i.e. settings where there is no adversarial pressure to reproduce the text. Human-written text is used to compare the extent to which the completions have exact matches in AuxDataset. The results show that unintended reproduction occurs more if the text is generated by models instead of humans, and if the task is expository, e.g. tutorials. Two system prompts are  investigated for mitigating unintended reproduction, yielding moderate success.

**Strengths:**

1) The paper is well-written and accessible. The figures effectively convey the key findings.
2) The empirical results are extensive and presented in a way that is easy to interpret. The authors test the most relevant models (including GPT, Claude, and Llama) and use varied datasets including real conversations.
3) The experiments are well-designed, the knowledge cutoffs and dataset limitations are addressed in the text.

**Weaknesses:**

1) Only exact matches are considered, excluding reproduction of close matches with a low hamming-distance or reproduction of semantics.
2) The results are harder to interpret due to the possibility that a large number of reproductions by both humans and models is not captured by using AuxDataset. Perhaps the extent of the problem could be estimated by running the tests on a dataset that is expanded with additional sources and comparing the resulting numbers to the current ones.
3) The selection of 50 character length seemed insufficiently motivated. Especially since it is different from the prior work and results in both memorised and unoriginal phrases being included.

**Questions:**

1) Are strings of length 40-60 (Line 45) or 50 (Line 129) considered?
2) Do the reproductions occur in similar contexts to the originals?
3) Lines 162-164 - what is the filtering procedure for the human-written text for it to not appear on the internet? If it is filtered, how did plagiarism appear in the human-written IMDb reviews?

Nitpicks:
1) Figure 1 could be improved by adjusting the colour scheme and ensuring the readability of the small text.
2) The use of “aligned” in section 5 could be more precise. While Christiano et al., 2017 and Ouyang et al., 2022 describe alignment as a continuous objective of RLHF fine-tuning, Nasr et al. (2023) simply uses “aligned” to describe models that have undergone RLHF. To avoid this ambiguity, more specific terms like “RLHF-tuned” or “alignment fine-tuned” could be used to describe these models.

---

> ### Author Response · Authors · 2024-11-21
>
> We thank the reviewer for their thoughtful suggestions and questions.
>
> > Only exact matches are considered, excluding reproduction of close matches with a low hamming-distance or reproduction of semantics.
>
> We agree that extending our study to a non-verbatim setting is interesting future work. However, we note that this is challenging due to the large cost of finding fuzzy matches in 10TB of internet data.
>
> > The results are harder to interpret due to the possibility that a large number of reproductions by both humans and models is not captured by using AuxDataset. Perhaps the extent of the problem could be estimated by running the tests on a dataset that is expanded with additional sources and comparing the resulting numbers to the current ones.
>
> AuxDataset indeed only provides a lower bound for reproduction. While we thank the reviewer for their constructive suggestion, we think there is no feasible approach: AuxDataset is already a 10TB snapshot of the Internet, and most publicly accessible sources overlap heavily with AuxDataset.
>
> > The selection of 50 character length seemed insufficiently motivated. Especially since it is different from the prior work and results in both memorised and unoriginal phrases being included.
>
> We provide a clearer motivation in our [general response](https://openreview.net/forum?id=590yfqz1LE&noteId=LBuTkeYE0Y) and the improved discussion in the paper. We are happy to provide further justification if desired. We also note that we report every possible number of characters in our distribution plots (e.g., the x-axis of Figure 3a).
>
> > Are strings of length 40-60 (Line 45) or 50 (Line 129) considered?
>
> We define *overlap rates* in terms of strings of length at least 50 characters. The updated paper avoids the misleading statement on Line 45.
>
> > Do the reproductions occur in similar contexts to the originals?
>
> This is a very interesting question. Unfortunately, AuxDataset does not allow us to retrieve the context of a snippet (efficiently) because its implementation is optimized for efficient lookups (see [Nasr et al., 2023](https://arxiv.org/abs/2311.17035) for details).
>
> > Lines 162-164 - what is the filtering procedure for the human-written text for it to not appear on the internet? If it is filtered, how did plagiarism appear in the human-written IMDb reviews?
>
> We source the human-written text from May 2024 or later, which ensures the text is neither in any model’s training data nor in AuxDataset. We do not perform any other form of filtering to avoid introducing biases. Plagiarism appears if, for example, an [IMDb review submitted in July 2024](https://www.imdb.com/review/rw9878463/) copies a [review from 2002](https://www.imdb.com/review/rw0349147/). The review from 2002 is in AuxDataset, hence the July 2024 review has a large verbatim match with AuxDataset.
>
> > Figure 1 could be improved by adjusting the colour scheme and ensuring the readability of the small text.
>
> We thank the reviewer for this input and updated Figure 1.

---

> ### Comment · Reviewer_VqkD · 2024-11-27
>
> Thank you for the response - it answered most of my questions, and I will raise my score.
>
> Re: original context of reproductions - I wonder whether these could be retrieved with the implementation from Nasr et al., 2023 by iteratively searching for all characters that could expand the current match.

---

> > ### Author Response · Authors · 2024-11-27
> > **Thank you and follow-up**
> >
> > We thank the reviewer for their follow-up and raising their score.
> >
> > > I wonder whether these could be retrieved with the implementation from Nasr et al., 2023 by iteratively searching for all characters that could expand the current match.
> >
> > This should indeed be possible, but is restricted to suffix context of a snippet. That is, since AuxDataset is implemented as a suffix array, we could efficiently expand a snippet to obtain a context that follows that snippet in AuxDataset. However, obtaining prefix context could be more expensive, since that likely requires a brute-force approach.

---

### Official Review · Reviewer_6ufj · 2024-11-04

**Soundness:** 3
**Presentation:** 4
**Contribution:** 4
**Rating:** 8
**Confidence:** 4

**Summary:**

The paper shows how language models can reproduce training data even with 'non-adversarial' prompts. While LLMs have been previously shown to reproduce training data, these experiments were conducted with adversarial assumptions, and the prompts used can be considered a violation of user policy by many LLM developers. The authors argue that even under the assumption of non-adversarial prompts, i.e., everyday use prompts that are not targeted at extracting training data, one can see LLMs regurgitating their training data. The authors provide a wide range of experiments on many different SOTA conversational LLMs and with many different categories of prompts to support their hypothesis.

**Strengths:**

- Incredibly relevant work. While the pessimist in me believes that LLM developers will always find new excuses to argue why LLMs regurgitating sensitive or proprietary data is not their responsibility, it is important to try and keep holding them accountable. In the context of adversarial reproduction of training data not being "typical or allowed user activity", this work plays an important role in highlighting how even everyday use of LLMs can reproduce training data.
- Wide range of experiments, both in terms of different models, as well as verifying various hypotheses. Lots of interesting insights.
- Qualitative analysis and handpicked examples. I was happy to see some qualitative analysis by the authors, especially of the long tail.

**Weaknesses:**

- The use of a 50-character limit for overlap rate. I'm not convinced that the 50-character limit is strong enough to cause issues for LLMs reproducing training data. I'm not familiar with legal precedence on reproducing text without attribution; but at least when quoting from other sources, the limits are usually looser - even the strictest being around 25-50 words and usually, it is a few hundred words (https://ogc.harvard.edu/pages/copyright-and-fair-use, https://stevelaube.com/how-much-can-i-quote-from-another-source-without-permission/). Although, it should be mentioned that the authors are very open about their overall results and also discuss the long-tailed part of the reproduction, which highlights some actual issues. But despite this, their main results and trends are focused on an overlap rate defined with a 50-character limit.
- Lack of details on additional prompts used in the experiments. The authors have created some manual human-written prompts, which are used alongside data scraped from Reddit, in their experiments. I understand that releasing all these prompts during the reviewing phase might not be practical, and I appreciate the authors mentioning that they will release them in a later version of the paper, but I would like to see some details in the paper to perform a proper review of their work. More details on this in the questions below.

**Questions:**

- Can the authors reason why the 50-character limit beyond simply the argument that non-adversarial prompts reproduce less training data? The qualitative analysis of those 50-character snippets is appreciated, but as the authors showed, many of them are common phrases that might not constitute problematic behaviour from LLMs.
- Can the authors provide more details on how their manual prompts were created? Were they crowdsourced, or written by the authors themselves? Were they sourced from how authors themselves commonly use LLMs, or were they thought up in bulk at once? Were there efforts made to categorize them into a variety of prompts (beyond the three broad categories used in the paper), or maybe efforts made to check this variety after the prompts were created? No answers are bad answers here, even if the prompts were written by the authors in bulk in one sitting to capture the broad categories defined, that's a good start. But in any case, details are needed.

---

> ### Author Response · Authors · 2024-11-21
>
> We thank the reviewer for their reinforcing feedback and thoughtful comments.
>
> > Can the authors reason why the 50-character limit beyond simply the argument that non-adversarial prompts reproduce less training data?
>
> We updated the paper with a clearer motivation for the 50-character threshold; see the [general response](https://openreview.net/forum?id=590yfqz1LE&noteId=LBuTkeYE0Y) for more details. We are happy to answer any follow-up questions.
>
> > I'm not convinced that the 50-character limit is strong enough to cause issues for LLMs reproducing training data.
>
> Besides the clearer motivation for 50+ character snippets, we also report distribution plots *for every possible character threshold* (e.g., Fig. 3 and 5). We further included a new Fig. 10 that contains details for every individual model.
>
> > Can the authors provide more details on how their manual prompts were created? Were they crowdsourced, or written by the authors themselves?
>
> If not mentioned otherwise, all prompts were created by the authors (manually). We updated App. A.1 and Table 1 to make this more explicit. Our updated submission also contains our code (including raw prompt data) and a link to the full dataset.
>
> > Were they sourced from how authors themselves commonly use LLMs, or were they thought up in bulk at once? Were there efforts made to categorize them into a variety of prompts (beyond the three broad categories used in the paper), or maybe efforts made to check this variety after the prompts were created?
>
> We started with the three text types (creative/expository/argumentative). We then brainstormed tasks that are i) based on real-world LLM usage (of the authors and other people on the internet), ii) covering a diverse set of prompts, iii) roughly equally distributed over text types. For each task, we invented concrete prompts in a batch. Our original set of prompts had task pairs that turned out to be very similar (e.g., Statement of Purpose and Motivation Letter); we replaced one instance of each pair to increase diversity/coverage.

---

> > ### Comment · Reviewer_6ufj · 2024-11-25
> >
> > The authors' responses are acknowledged. Thank you for clarifying the details further.
> >
> > I'll stick with my already positive score.

---

### Official Review · Reviewer_5QLf · 2024-11-04

**Soundness:** 3
**Presentation:** 2
**Contribution:** 3
**Rating:** 5
**Confidence:** 4

**Summary:**

The paper tackles the problem of mitigating non-adversarial training data reproduction (natural and benign prompts from users revealing training data verbatim) in LLMs. The experiments find that in some cases as much as 15% of text output by LLMs overlaps with moderate snippets (40-60 characters) of the Internet. A comparative analysis finds that human-written text has far less overlap compared to LLM generated text. The paper proposes prompting strategies to close this human and LLM gap. Though these strategies close the gap between LLMs and humans on average, the paper suggests that worst-case reproduction might need stronger adversarial defenses.

The classes of tasks chosen for LLM text generation in this paper can be broadly classified into *creative writing*, *expository writing*, and *argumentative writing*. Since training data information is not available for certain models, the training data is approximated by collecting a large dataset of Web content (AUXDATASET).

The primary metric used is overlap rate (the percentage of characters in each generation that belong to a substring of at least 50-consecutive characters found exactly in AUXDATASET).

**Strengths:**

The paper is an extensive analysis of situations where LLMs generate text from training data verbatim, even when not explicitly prompted to reveal such information. The later case has been seen in adversarial attacks against LLMs in recent research. So, the results from this study can be used to inform us about scenarios where data leakage happens without explicit adversarial effort.

**Weaknesses:**

The current text is ordered as a set of subsections with a verbatim description of experimental steps. The presentation lacks focus on the main contributions of this research. For example, if this is the case, it should probably be highlighted that LLM data leakage studies for benign prompts haven't been looked at. Furthermore, instead of presenting all the results (as in Section 3) as small headings and text, it would help to have an additional small section which highlights the most important contributions which readers can take away from the paper.

I have some concerns about the collection of human data regarding plagiarism and contamination. Please refer to the Questions section.

**Questions:**

1. (Line 162 - Question about human-written baseline) Even with this measure of choosing content after LLM cut-off date and content which is not part of AUXDATASET, how is it confirmed that the content taken from Reddit is not LLM generated? Is it not possible that an LLM might have been used to generate it?

2. (Line 321) The paper mentions, “We find that LLMs reproduce more existing data than humans, except when humans do blatant plagiarism.” I might have missed it in the text, but it would be great to have some clarification regarding how this is controlled? For example, given a prompt like “Write a tutorial about setting up an Nginx server.”, humans might be prone to copy data verbatim from top-ranked articles on a search engine. There is a discussion in Line 404 about IMDb reviews, but what measures were taken for Reddit content?

---

> ### Author Response · Authors · 2024-11-21
>
> We thank the reviewer for their thoughtful comments.
>
> > The current text is ordered as a set of subsections with a verbatim description of experimental steps. The presentation lacks focus on the main contributions of this research.
>
> We have updated the paper to make our main focus more clear; please refer to the [general response](https://openreview.net/forum?id=590yfqz1LE&noteId=LBuTkeYE0Y) for details. We are happy to answer any remaining ambiguities. Finally, we note that each paragraph in Sec. 3 and 4 corresponds to one finding of our study.
>
> > Even with this measure of choosing content after LLM cut-off date and content which is not part of AUXDATASET, how is it confirmed that the content taken from Reddit is not LLM generated? Is it not possible that an LLM might have been used to generate it?
>
> In short, even if a few rare instances of human-written baselines contain some LLM-generated text, this does not weaken our comparison (it might even make it stronger). We discuss this more in the [general response](https://openreview.net/forum?id=590yfqz1LE&noteId=LBuTkeYE0Y).
>
> > The paper mentions, “We find that LLMs reproduce more existing data than humans, except when humans do blatant plagiarism.” I might have missed it in the text, but it would be great to have some clarification regarding how this is controlled? For example, given a prompt like “Write a tutorial about setting up an Nginx server.”, humans might be prone to copy data verbatim from top-ranked articles on a search engine. There is a discussion in Line 404 about IMDb reviews, but what measures were taken for Reddit content?
>
> We can indeed not rule out plagiarism in human-written responses for the Reddit content (WritingPrompts and ELI5 tasks). However, we did perform a manual investigation (as for IMDb reviews), which did not reveal any significant cases of blatant plagiarism (as opposed to IMDb reviews). This matches our Fig. 5a/c, where human texts contain fewer reproduced snippets of any length compared to LLMs, and Fig. 6 where the mean overlap rate for humans on creative/expository tasks is below 2% (and the median is 0).

---

> > ### Comment · Reviewer_5QLf · 2024-11-26
> >
> > Thank you for the response. Could you please provide clarification about this sentence “However, we did perform a manual investigation (as for IMDb reviews), which did not reveal any significant cases of blatant plagiarism (as opposed to IMDb reviews).”  It seems like a typo. Is "for" and "opposed" to being used correctly here?

---

> > > ### Author Response · Authors · 2024-11-26
> > > **Clarification on sentence**
> > >
> > > We apologize for the slightly weird phrasing. In other words:
> > >
> > > We manually investigated all three human baselines (IMDb reviews, WritingPrompts, ELI5) regarding blatant human plagiarism. We did find a lot of plagiarism for IMDb reviews, but we did not find any instances of blatant human plagiarism for WritingPrompts or ELI5.

---

### Official Review · Reviewer_sm5N · 2024-11-11

**Soundness:** 3
**Presentation:** 4
**Contribution:** 3
**Rating:** 6
**Confidence:** 5

**Summary:**

This work discusses the case of unintentional reproduction of training data by large language models. While most of the literature discusses an adversarial nature of prompting to extract training data, this work tries to quantify how frequently this influence happens in a non-adversarial situation. One of the findings of the work is that non-adversarial reproduction is much higher in expository tasks than in creative ones, and even prompting techniques, while they can reduce the average reproduction rate, are not sufficient to prevent longer sequences from appearing. One of the highlight results of this work is that about 15% of the text output by popular conversation language models overlaps with short snippets of text on the internet, much higher than baseline rates by humans on the same task.

**Strengths:**

Overall, this paper was a joy to read. I found it to be very thoughtfully written. I routinely ended up in situations where I had a particular thought experiment in mind, and the next section showed just that set of ablations or experiments.

- I liked Figure 4(b) which shows how reproduction strongly depends on the task. This consolidates an important finding/hypothesis. At a high level, while the paper reports that 8-15% of LLM-generated text overlaps with existing online snippets, it goes further to analyze the types of overlaps. This also highlights the complexity of defining "problematic" reproduction.
- I found the experiment on extracting Quotations quite interesting, in particular, because it shows incorrect attribution of the quote.
- Distribution of overlap lengths: A small percentage of outputs contain very long reproduced sequences. This long-tail phenomenon suggests that even with low average reproduction rates, LLMs can still pose risks in specific cases.

**Weaknesses:**

- Frequency of Reproduced Sequences: The paper could benefit from clarifying how often the reproduced sequences appear within their training data proxy, AUXDATASET. Understanding whether these snippets are rare or commonly encountered would help contextualize the reproduction risks.

- Justification of 50-Character Threshold: The choice of a 50-character threshold to define reproduced sequences is not fully justified. In particular, this is quite different from past work. While some examples in the Appendix suggest that 50 characters is a meaningful number, I believe most examples highlight that such sequence lengths can be so common in the natural language distribution that their reproduction does not matter. Further explanation would help readers assess whether this threshold adequately captures the difference between common phrases and more problematic reproductions.

- Data in Figure 2(b): Figure 2(b) appears to have only partial bar plots for some models (Llama and GPT), making the comparison across models less robust. Or am I missing something here?

Overall, I am constantly battling between thinking that 50 characters is too less, and then seeing the argument that these reproduction rates are much higher than humans. This makes me wonder if humans are the right baseline here. Would a human with a passage (RAG style reading comprehension) be a better baseline? There is a qualitative dichotomy here: the 50 characters do not feel meaningful when visualized, yet stay higher than what a human would reproduce.

**Questions:**

Please refer to sections above and answer the questions

---

> ### Author Response · Authors · 2024-11-21
>
> We thank the reviewer for their kind words and constructive comments.
>
> > Frequency of Reproduced Sequences: The paper could benefit from clarifying how often the reproduced sequences appear within their training data proxy, AUXDATASET. [...]
>
> This would be very interesting. Unfortunately, there is no way to reliably estimate the frequency of snippets in any model’s training data. First, model providers might perform deduplication of their training data, such that a snippet’s frequency on the internet might not match its frequency in the training data. Second, AuxDataset combines potentially overlapping sources (e.g., multiple copies of Wikipedia in different datasets). In both instances, we can reliably approximate whether a snippet is part of the data, but not with which frequency.
>
> > Justification of 50-Character Threshold: The choice of a 50-character threshold to define reproduced sequences is not fully justified. [...]. Further explanation would help readers assess whether this threshold adequately captures the difference between common phrases and more problematic reproductions.
>
> We hope the explanation in our [general response](https://openreview.net/forum?id=590yfqz1LE&noteId=LBuTkeYE0Y
> ) and the improved discussion in the paper provide a better justification. We are happy to answer follow-up questions.
>
> > Data in Figure 2(b): Figure 2(b) appears to have only partial bar plots for some models (Llama and GPT), making the comparison across models less robust. Or am I missing something here?
>
> This is correct and a consequence of our post-hoc analysis of WildChat and LMSYS-Chat-1M. For example, WildChat consists solely of ChatGPT conversations and thus only has generations for GPT models. This is okay; the goal of Fig. 2(b) is to highlight that non-adversarial reproduction also happens in the wild (outside of our controlled experiments), not to provide a robust comparison between models.
>
> > Overall, I am constantly battling between thinking that 50 characters is too less, and then seeing the argument that these reproduction rates are much higher than humans. [...]. Would a human with a passage (RAG style reading comprehension) be a better baseline? [...]
>
> We thank the reviewer for this in-depth thought; however, there are subtleties that break this analogy. For LLMs, we explicitly focus on *copying/reproduction from the training data*, not from the prompt. However, humans copying text from a provided passage corresponds to LLMs copying from their prompt/context. A better analogy are humans who are able to recall and lookup information from memory—which is exactly what we measure.

---

> > ### Comment · Reviewer_sm5N · 2024-12-03
> >
> > Dear Authors,
> >
> > Thank you for your thoughtful responses to my review. Your paper has sparked important discussions about the nature of text reproduction in language models, and I appreciate your careful attention to the concerns raised.
> >
> > **Training Data Frequency Analysis**
> > This is not hard for open datasets. Check out: https://huggingface.co/spaces/liujch1998/infini-gram
> >
> > **50-Character Threshold and User Perception**
> > The threshold choice remains a crucial concern that deserves more rigorous investigation. This is particularly important given the sensitive nature of text reproduction and its implications for content creators and users. I strongly recommend:
> > - Conducting user studies to understand how different stakeholders (content creators, users, domain experts) perceive text reproduction at various lengths
> > - Including controlled experiments where participants evaluate the "problematic" nature of reproduced text across different thresholds and contexts
> > - Incorporating findings from the NLP community's established methodologies for human evaluation and annotation
> >
> > The NLP community has a rich history of using human studies to validate important thresholds and metrics. This would be an excellent opportunity to bring that tradition to this important problem rather than relying solely on computational justifications.
> >
> > **Figure 2(b) Clarification**
> > Thank you for explaining the partial nature of the bar plots. To prevent confusion can you add a note in the figure caption explaining the data availability?
> >
> > **Human Baseline Comparison**
> > Your clarification about the distinction between prompt copying and training data reproduction is well-taken. To strengthen this point:
> > - Consider adding a diagram illustrating the different types of "copying" (human memory vs. LLM training vs. RAG)
> > - Include a discussion of how different memory mechanisms might affect reproduction patterns
> >
> >
> >
> > Your paper makes important contributions to understanding non-adversarial reproduction in language models, and your responses have clarified several key points. The methodological choices, while sometimes difficult to justify perfectly, seem reasonable given the constraints of studying this phenomenon. I particularly appreciate the thorough analysis of task-dependent reproduction rates and the careful consideration of what constitutes "problematic" reproduction.

---

> > > ### Author Response · Authors · 2024-12-03
> > > **Thank you and follow-up**
> > >
> > > We thank the reviewer for taking the time to read our rebuttal and providing further constructive feedback.
> > >
> > > **Training Data Frequency Analysis.**
> > > We agree that it's not difficult from a technical perspective, and useful for open datasets. The point is that it does not provide meaningful information for *private/unknown datasets*. Our frequency estimates would not necessarily represent any relevant information about the private training data because (1) model providers often deduplicate and (2) open-source datasets often overlap (e.g., they all contain Wikipedia).
> > >
> > > **Human studies.**
> > > We agree this is an interesting and meaningful additional direction, but it falls outside the scope of our work. We will mention this more explicitly in the future work section.
> > >
> > > **Figure 2(b) Clarification.**
> > > Note that the caption already reads "Notice that not all models exist in both datasets". However, we will think about making this more explicit.
> > >
> > > **Human Baseline Comparison.**
> > > We thank the reviewer for their constructive comments. However, note that we already discuss blatant plagiarism and use human baselines to motivate that the snippets we consider are unlikely to be generated by humans. Discussing the details of each memory mechanism would require additional empirical evidence and thus falls outside the scope of our work.

---

### Author Response · Authors · 2024-11-21
**General response to all reviewers**

We thank all reviewers for their time and constructive feedback. In this general response, we expand on two points multiple reviewers raised: the 50-character threshold and potential contamination of human baselines. We further uploaded a revision with changes summarized as follows:

1. As promised, we released all data (prompts, LLM completions, and matches with AuxDataset) via [https://huggingface.co/datasets/nonadvreproduction/paper-anon-data](https://huggingface.co/datasets/nonadvreproduction/paper-anon-data) and included our code as supplementary material.
2. We included additional models from the Gemini 1.5 and OpenAI o1 families to provide an even broader picture.
3. To better motivate and justify the 50-character threshold for overlap rates, we added clarifications and additional intuition throughout the paper. We encourage reviewers to check the new introduction and motivation sections.
4. Relatedly, we included the fraction of text containing a reproduced snippet *for every possible reproduction threshold and model* as Figure 10 in the appendix.

**50-character threshold:** Many reviewers wonder whether the minimum snippet length of *at least 50 characters* is meaningful. First, we note that the distribution plots (Fig. 3a, 5, 7b, and the new Fig. 10\) show how much text contains a reproduced snippet *for every possible threshold* (x-axis values). Hence, we do *not just* report reproduction of ≥50-character snippets, but use a threshold primarily to provide high-level quantitative results.

Second, we choose the particular 50-character value to bridge existing studies of linguistic novelty (short n-grams, \<10 characters) and long adversarial extraction (\>200 characters). We find 50 characters to be an interesting sweet spot where reproduction transitions from common idioms/expressions to problematic reproduction of training data. We updated the paper to elaborate more on those nuances.

**Human baselines being LLM-generated text:** Some reviewers (justifiably) wonder if our human baselines contain LLM-generated text. We cannot reliably detect or control for this. However, even if our baselines would contain a small fraction of LLM-generated text, we argue that this makes our results even stronger\! Imagine we could reliably detect all LLM-generated text in our human baselines. This text will have similar overlap rates as our LLM generations. But those rates are higher than what we already measure for humans. Hence, if we rid our human baselines from LLM-generated text, the human overlap rates would be the same or even lower, and the *gap between humans and LLMs even larger*.

---

### Meta-Review · Area_Chair_wkLq · 2024-12-17

**Metareview:**

This paper thoroughly studies non-adversarial memorization behaviors in LLMs.  The reviewers largely vote for acceptance, and I agree with the reviewers.  The reviewers raised a number of concerns which did not mitigate their positive overall feelings.  For example, multiple reviewers brought up the choice of 50 characters for the threshold, but the authors pointed out that their plots are not at all restricted to this threshold choice, and the threshold was just employed for illustrative purposes.  Reviewers also asked the authors to compute frequency of reproduced sequences in the training data, but this is not known since people preprocess their data, often deduplicating.  Reviewers mentioned that text from various sources, for example Reddit, could be LLM generated, which indeed is a possible confounder, but it does not invalidate the results and is an interesting topic for the future.  Similarly, I do not think it is a problem that the paper focuses on only exact matches although I do think that is a limitation that should be explored in future work.  Finally, the authors have added further experimental details which address a number of points raised by reviewers.  Overall, the feedback does not look disqualifying to me, and the reviewers and I broadly view this paper favorably.

**Additional Comments On Reviewer Discussion:**

The authors comprehensively addressed reviewer feedback, including many new experiments and paper edits, and they also published their data.

---

### Decision · Program_Chairs · 2025-01-22

Accept (Poster)